# Selective molecular impairment of spontaneous neurotransmission modulates synaptic efficacy

Devon C. Crawford[1],*, Denise M.O. Ramirez[2],*, Brent Trauterman[1], Lisa M. Monteggia[1] & Ege T. Kavalali[1,3]

Recent studies suggest that stimulus-evoked and spontaneous neurotransmitter release processes are mechanistically distinct. Here we targeted the non-canonical synaptic vesicle SNAREs Vps10p-tail-interactor-1a (vti1a) and vesicle-associated membrane protein 7 (VAMP7) to specifically inhibit spontaneous release events and probe whether these events signal independently of evoked release to the postsynaptic neuron. We found that loss of vti1a and VAMP7 impairs spontaneous high-frequency glutamate release and augments unitary event amplitudes by reducing postsynaptic eukaryotic elongation factor 2 kinase (eEF2K) activity subsequent to the reduction in N-methyl-D-aspartate receptor (NMDAR) activity. Presynaptic, but not postsynaptic, loss of vti1a and VAMP7 occludes NMDAR antagonist-induced synaptic potentiation in an intact circuit, confirming the role of these vesicular SNAREs in setting synaptic strength. Collectively, these results demonstrate that spontaneous neurotransmission signals independently of stimulus-evoked release and highlight its role as a key regulator of postsynaptic efficacy.

[1] Department of Neuroscience, UT Southwestern Medical Center, Dallas, Texas 75390, USA. [2] Whole Brain Microscopy Facility, Department of Neurology and Neurotherapeutics, UT Southwestern Medical Center, Dallas, Texas 75390, USA. [3] Department of Physiology, UT Southwestern Medical Center, Dallas, Texas 75390, USA. * These authors contributed equally to this work. Correspondence and requests for materials should be addressed to E.T.K. (email: Ege.Kavalali@UTSouthwestern.edu).

Neurotransmitter release exerts a strong long-term influence on the sensitivity of postsynaptic targets to presynaptic inputs. In classic experiments, denervation has been shown to render target tissues supersensitive to neurotransmitters via upregulation of postsynaptic receptors[1]. Similar homeostatic mechanisms are known to function at mammalian central synapses, where chronic activity blockade induces an increase in miniature excitatory postsynaptic current (mEPSC) amplitudes[2]. Recent evidence suggests a key role for spontaneous neurotransmitter release in this process, as inhibition of postsynaptic receptors or neurotransmitter depletion from spontaneously recycling vesicles after suppression of activity increases postsynaptic sensitivity[3–5]. However, it is unclear if spontaneous neurotransmission acts independently of changes in activity[6] or is part of a continuum of neurotransmitter release modes[7,8]. To address whether spontaneous neurotransmitter release has a specific, independent role in setting postsynaptic efficacy requires selective inhibition of this resting release process without significant alterations in neuronal activity.

Recent work has identified distinct molecular pathways that selectively mediate spontaneous release independently of evoked neurotransmission[6,9]. Soluble N-ethylmaleimide-sensitive-factor attachment protein receptors (SNAREs) that function in endosomal trafficking pathways such as Vps10p-tail-interactor-1a (vti1a) and vesicle-associated membrane protein 7 (VAMP7; also known as tetanus toxin-insensitive vesicle-associated membrane protein or TI-VAMP) function in intracellular trafficking pathways but also reside on synaptic vesicles[10–12], where they have dedicated roles in spontaneous neurotransmission[13–16]. Vti1a-containing vesicles are distinct from those containing the canonical vesicular SNARE synaptobrevin2 and preferentially traffic at rest, maintaining the high-frequency component of spontaneous neurotransmission[15]. VAMP7-containing vesicles are also preferentially released at rest[14,15] but mobilize in response to the secreted glycoprotein Reelin[13]. Recent evidence has also shown that spontaneous neurotransmission signals to a segregated pool of postsynaptic receptors[17], thereby giving this form of neurotransmission the potential to induce downstream signalling cascades, such as postsynaptic phosphorylation of eukaryotic elongation factor 2 (eEF2), that are distinct from those activated via action potential-evoked neurotransmission[18,19]. These findings raise important new questions regarding whether these different forms of neurotransmitter release, driven by molecularly distinct synaptic vesicle populations, independently trigger specific postsynaptic responses.

In this study, we selectively impaired spontaneous release without substantially affecting basal action potential-evoked excitatory neurotransmission by knocking down the non-canonical vesicular SNAREs vti1a and VAMP7 in hippocampal neurons. We report that loss of vti1a and VAMP7 impairs spontaneous glutamate release and elicits synaptic scaling of α-amino-3-hydroxy-5-methyl-4-isoxazolepropionic acid (AMPA)-mEPSC amplitudes through postsynaptic eEF2 signalling. By the selective reduction of vti1a and VAMP7 in subpopulations of neurons, we also determined that presynaptic, rather than postsynaptic, loss of vti1a and VAMP7 induces synaptic scaling and impairs N-methyl-D-aspartate receptor (NMDAR) antagonist-induced synaptic potentiation, a form of plasticity associated with fast-acting antidepressant responses[3,18]. Collectively, our results strongly suggest that vti1a and VAMP7 function presynaptically to promote spontaneous neurotransmission and maintain basal levels of postsynaptic efficacy in a manner independent from synchronously evoked neurotransmission.

## Results

### Vti1a and VAMP7 loss impairs spontaneous neurotransmission.
Synaptic vesicle proteins vti1a[15] and VAMP7 (refs 13–15) have

been previously shown to be involved in spontaneous fusion. In order to assess the impact of these vesicular SNAREs on synaptic efficacy, we cultured rat hippocampal neurons and used vti1a and VAMP7 short hairpin RNA (shRNA) expression via lentivirus infection to reduce vti1a and VAMP7 protein levels (Supplementary Fig. 1). We used a double knockdown (DKD) strategy to reduce expression of these proteins and avoid potential functional redundancy of vti1a and VAMP7 in spontaneous neurotransmission. We first measured uptake of an antibody directed against the luminal domain of synaptotagmin 1 under resting conditions (Fig. 1a). The uptake of luminal synaptotagmin antibodies is a well-validated measure of synaptic vesicle recycling[20,21]. Vti1a/VAMP7 DKD neurons showed 29% less antibody uptake than control neurons (Fig. 1b), indicative of a reduction in spontaneous synaptic vesicle recycling. We next performed whole-cell patch clamp recordings of AMPA-mEPSCs in control and vti1a/VAMP7 DKD neurons in the presence of tetrodotoxin (TTX) to block action potentials (Fig. 1c). Under these conditions, we detected a significant shift in AMPA-mEPSC inter-event intervals towards longer periods in the vti1a/VAMP7 DKD neurons (Fig. 1d), but average frequency data only trended towards a decrease (control: $7.25 \pm 1.24$ Hz; vti1a/VAMP7 DKD: $5.68 \pm 0.74$ Hz; $P = 0.27$), suggesting that high frequency events may be selectively targeted. The increase in inter-event intervals was rescued by co-expression of shRNA-resistant vti1a, and vti1a overexpression by itself increased AMPA-mEPSC frequency (Supplementary Fig. 2a,b,d). No change in synaptic density was observed in vti1a/VAMP7 DKD neurons (control: $415 \pm 36$ synaptotagmin 1 puncta per field; vti1a/VAMP7 DKD: $497 \pm 54$ synaptotagmin 1 puncta per field; $n = 12$ images from three independent cultures; $P = 0.22$), suggesting that this reduction in AMPA-mEPSC frequency is driven by reduced presynaptic vesicle fusion rather than by altered synaptic development.

As loss of vti1a and VAMP7 appeared to selectively target the high frequency (short inter-event interval) component of mEPSCs (Fig. 1d), we measured the frequency of AMPA-mEPSC bursts, which are representative of multi-quantal glutamate release[22,23]. Bursts of AMPA-mEPSCs were defined as groups ($\geq 2$) of consecutive events with inter-event intervals $< 40$ ms and were reliably observed throughout both control and vti1a/VAMP7 DKD recordings (Fig. 1e); however, a significant decrease in AMPA-mEPSC burst frequency was observed in vti1a/VAMP7 DKD neurons (Fig. 1f). Additionally, when these bursts were analysed in isolation, the average frequency of AMPA-mEPSCs decreased significantly after loss of vti1a and VAMP7 (control: $1.08 \pm 0.23$ Hz; vti1a/VAMP7 DKD: $0.54 \pm 0.11$ Hz; $P = 0.036$); however, when bursts were removed from the data set, the remaining AMPA-mEPSC frequency did not significantly decrease (control: $6.34 \pm 1.03$ Hz; vti1a/VAMP7 DKD: $5.14 \pm 0.64$ Hz; $P = 0.32$). Altogether, these results suggest that vti1a and VAMP7 drive high-frequency bursts of AMPA-mEPSCs.

### NMDAR activation by vti1a and VAMP7 mediated release.
Rapid release of glutamate quanta at rest would be expected to increase the probability of NMDAR activation and calcium entry and in turn affect downstream postsynaptic signalling cascades[6,19,22]. Thus, we tested the hypothesis that defective high-frequency glutamate release in vti1a/VAMP7 DKD neurons affects spontaneous NMDAR activity. NMDAR-mEPSCs were recorded at rest in the presence of glycine and the absence of $Mg^{2+}$ (Fig. 2a,b). The use-dependent and slowly reversible NMDAR blocker MK-801 (refs 17,24,25) was applied for 1 min before a 90 s washout. This washout prevented the measurement

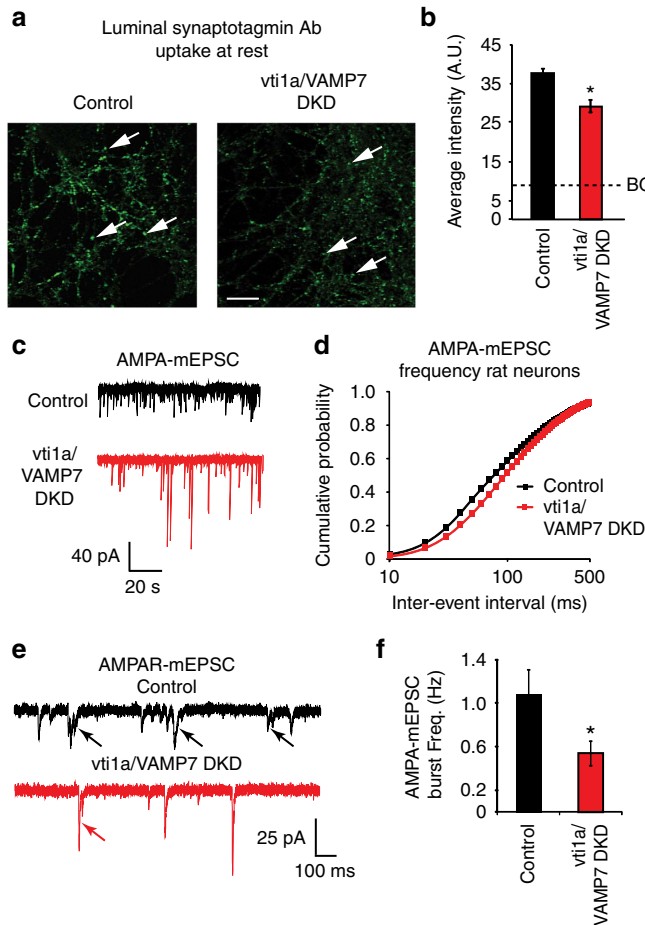

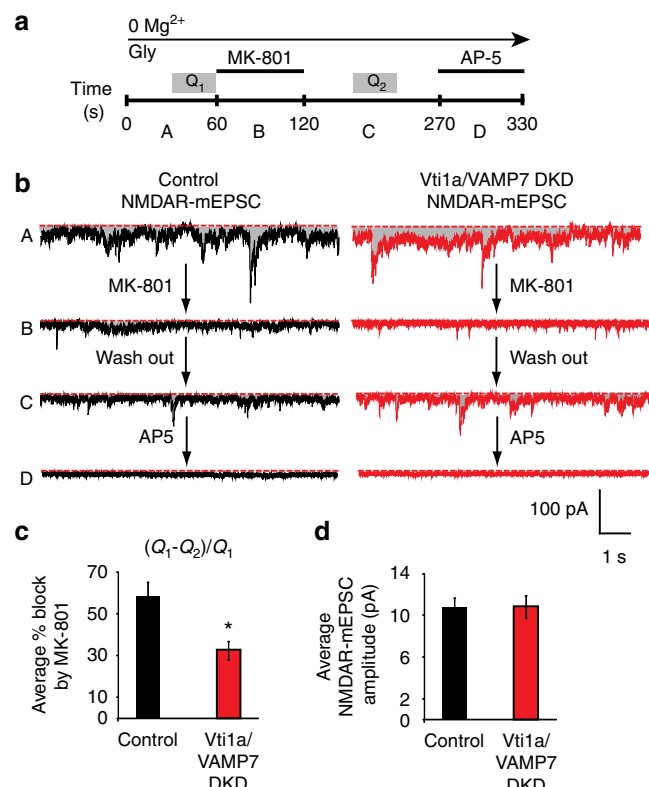

**Figure 1 | Vti1a and VAMP7 loss in cultured rat hippocampal neurons decreases synaptic vesicle trafficking at rest.** (**a**) Representative images of anti-luminal domain of synaptotagmin 1 antibody uptake at rest in control and vti1a/VAMP7 DKD (DKD) neurons. White arrows indicate representative areas selected for intensity analysis typical of presynaptic boutons. Scale bar, 10 μm. (**b**) Quantification of anti-luminal domain of synaptotagmin 1 antibody intensity at rest in control and vti1a/VAMP7 DKD neurons (control: $n = 167$ puncta from 10 images from two independent cultures; vti1a/VAMP7 DKD: $n = 221$ puncta from 11 images from two independent cultures; $P = 0.002$). Background (BG; dashed line) fluorescence levels were obtained from similar experiments in which the primary antibody was omitted. (**c**) Representative traces of spontaneous AMPA event (AMPA-mEPSC) recordings in control and vti1a/VAMP7 DKD neurons. (**d**) Cumulative probability histograms of AMPA-mEPSC inter-event intervals (control: $n = 29$ recordings from seven independent cultures; vti1a/VAMP7 DKD: $n = 31$ recordings from seven independent cultures; Kolmogorov–Smirnov test; $P = 0.0001$). (**e**) Representative traces of AMPA-mEPSC recordings containing mini-bursts (indicated by arrows) in control and vti1a/VAMP7 DKD neurons. (**f**) Average AMPA-mEPSC burst frequency was significantly decreased in vti1a/VAMP7 DKD neurons compared with control in the same recordings analysed in **d** ($P = 0.036$).

**Figure 2 | Vti1a- and VAMP7-contaning synaptic vesicles regulate tonic NMDA receptor (NMDAR) activation.** (**a**) Diagram showing the experimental strategy. Basal NMDAR-mEPSCs were recorded from cultured rat hippocampal neurons in the absence of extracellular $Mg^{2+}$ and the presence of the co-agonist, glycine (10 μM). Neurons were then incubated with the use-dependent and slowly reversible NMDAR antagonist MK-801 (10 μM) for 1 min and subsequently washed for 1.5 min. AP-5 (50 μM) was added at the end of the experiment to confirm that the mEPSCs originated from NMDARs. Q1 and Q2 indicate regions where total charge transfer was measured and used for analysis of percent NMDAR blockade by MK801. A-D corresponds to regions containing the sample traces shown in **b**. (**b**) Sample NMDAR-mEPSC traces from control and vti1a/VAMP7 DKD neurons subjected to the indicated treatments shown in panel **a**. (**c**) Quantification of percent block of the NMDAR by MK-801 in control and vti1a/VAMP7 DKD neurons. MK-801 was significantly less effective in blocking NMDAR activity in vti1a/VAMP7 DKD neurons compared with control (control: $n = 11$ recordings from three independent cultures; vti1a/VAMP7 DKD: $n = 9$ recordings from three independent cultures; $P = 0.01$). (**d**) Average basal NMDAR-mEPSC event amplitudes were unchanged in vti1a/VAMP7 DKD neurons compared with control neurons from data analysed in **c** ($P = 0.97$).

of any non-use-dependent block of the receptor and should not allow substantial receptor recovery on the timescale of this experiment[25,26]. The specific, non-use-dependent NMDAR antagonist AP-5 was then added at the end of the experiment to confirm that the mEPSCs originated from NMDARs. As MK-801 only blocks open NMDARs, the extent of NMDAR block in the presence of MK-801 provides a reliable estimate of the NMDAR activity per synapse, independent of the number of synapses[17,25].

The total charge transfer as a cumulative measure of NMDAR activity was quantified from a 30-s period before and after MK-801 application ($Q_1$ and $Q_2$ in Fig. 2a). The remaining activity after MK-801 was then subtracted from the baseline activity to calculate the amount of activity blocked by the treatment, and this number was then divided by the baseline activity to calculate the percentage of activity blocked by MK-801 (Fig. 2c). MK-801 treatment itself significantly reduced NMDAR-mEPSC activity in both control and vti1a/VAMP7 DKD neurons, consistent with earlier results from our group[17]; however, vti1a/VAMP7 DKD neurons retained more activity than control neurons during the washout phase and, therefore, experienced reduced block (Fig. 2b,c). Under these

conditions, the percentage block of NMDARs at rest by MK-801 was significantly reduced in vti1a/VAMP7 DKD neurons compared with control (Fig. 2c). Because MK-801 is a use-dependent blocker, this is consistent with our hypothesis that the multi-quantal glutamate release mediated by vti1a- and VAMP7-containing vesicles normally activates NMDARs under resting conditions. With fewer high-frequency bursts in vti1a/VAMP7 DKD neurons, NMDARs may be less activated by spontaneous neurotransmission and, therefore, less blocked by MK-801. We also observed that AP-5 treatment blocked all activity in both control and vti1a/VAMP7 DKD neurons, indicating the specificity of the recording conditions (Fig. 2b). Additional analysis of clearly identifiable NMDAR-mEPSC events revealed that their amplitudes did not differ between control and vti1a/VAMP7 DKD neurons (Fig. 2d). Altogether, these results suggest that loss of vti1a and VAMP7 impairs high-frequency spontaneous glutamate release that normally maintains resting NMDAR activation.

**Loss of vti1a and VAMP7 does not affect network activity**. We have previously shown that knockdown of either vti1a or VAMP7 alone does not affect evoked synaptic transmission[13,15]. To assess the possibility that vti1a/VAMP7 DKD and the resultant decrease in spontaneous event frequency affects global background activity, we recorded endogenous network activity in the presence of the $\gamma$-aminobutyric acid (GABA)$_A$ receptor antagonist picrotoxin (Fig. 3a). Control and vti1a/VAMP7 DKD neurons exhibited similar levels of excitatory background activity when analysed as overall charge transfer (Fig. 3b) or as burst number (Fig. 3c). Supporting this observation, similar action potential frequencies, burst frequencies, and action potentials per burst were observed between control and vti1a/VAMP7 DKD neurons in current-clamp mode in the absence of receptor blockers (Supplementary Fig. 3). These data show that although vti1a and VAMP7 control a portion of spontaneous neurotransmitter release, they do not significantly affect endogenous network activity.

**Loss of vti1a and VAMP7 induces excitatory synaptic scaling**. Although AMPA-mEPSC frequency is reduced after vti1a/ VAMP7 DKD, AMPA-mEPSC amplitude is increased compared with control neurons (Figs 1c and 3d–f). These effects could be rescued by co-expression of shRNA-resistant vti1a, and overexpression of vti1a alone in control neurons significantly decreased AMPA-mEPSC amplitude (Supplementary Fig. 2c,e). A rank-order plot of AMPA-mEPSC event amplitudes indicates a multiplicative increase of 70% in vti1a/VAMP7 DKD neurons compared with control (Fig. 3f), consistent with homeostatic synaptic scaling[27]. We also found that this increase in AMPA-mEPSC amplitude is similar to the increase measured after pharmacological NMDAR block, a technique known to produce synaptic scaling[4,28], and NMDAR block did not produce further scaling in vti1a/VAMP7 DKD neurons (Supplementary Fig. 4a,b), further suggesting that loss of vti1a and VAMP7 likely produce scaling via reduced NMDAR signalling. In parallel experiments, we recorded GABA miniature inhibitory postsynaptic currents (GABA-mIPSCs) in control and vti1a/VAMP7 DKD neurons. Consistent with the results from AMPA-mEPSCs, a reduction in vti1a and VAMP7 significantly decreased GABA-mIPSC frequency; however, in contrast to excitatory events, no significant change in GABA-mIPSC amplitudes was observed (Supplementary Fig. 5). Taken together, these results suggest synaptic scaling is selectively induced at glutamatergic synapses after loss of vti1a- and VAMP7-driven vesicle fusion.

To determine whether scaling of AMPA-mEPSC amplitudes in response to quantal release events is exclusive to spontaneous neurotransmission, we next recorded asynchronous evoked release in the presence of SrCl$_2$ and the absence of CaCl$_2$ to isolate unitary currents in control and vti1a/VAMP7 DKD neurons. Asynchronous unitary events evoked in the presence of Sr$^{2+}$ are thought to represent action potential-evoked neurotransmitter release that would have been synchronous in the presence of Ca$^{2+}$ (refs 29–32), so this method allowed us to measure quantal events from this mechanism of synaptic release. We found that evoked asynchronous unitary AMPA-EPSC amplitudes were augmented in vti1a/VAMP7 DKD neurons (Fig. 3g). Rank-order plots of the evoked asynchronous unitary AMPA-EPSC amplitudes indicates a multiplicative increase of 64% (Fig. 3h), closely resembling the amount of synaptic scaling of spontaneous AMPA-mEPSC events (Fig. 3f). Evoked bulk AMPA-EPSC amplitudes in Sr$^{2+}$ were also non-significantly increased (control: $1.11 \pm 0.27$ nA; vti1a/VAMP7 DKD: $1.70 \pm 0.13$ nA; $n = 4$ recordings from two independent cultures; $P = 0.06$), although de-synchronized excitatory response amplitudes in dissociated cultures are variable. These results indicate a global increase in synaptic efficacy, but not necessarily synchronous action potential-driven activity, upon loss of vti1a and VAMP7.

**Downstream eEF2 kinase activity maintains synaptic efficacy**. The data presented confirm previous observations from our laboratory and others that vti1a[15] and VAMP7 (refs 13–16) are specifically involved in spontaneous neurotransmission and, importantly, extend these findings to show that these vesicles participate in rapid glutamate release at rest and signal through NMDARs to regulate synaptic efficacy. In earlier work, it has been shown that spontaneous neurotransmission-driven NMDAR activity signals through eEF2 kinase (eEF2K) to control local dendritic protein translation[19,33], and phosphorylated eEF2 loses its affinity for the ribosome, slowing or stopping protein synthesis[34]. It is noteworthy that inhibition of eEF2K signalling downstream of NMDAR activation at rest has been implicated in NMDAR antagonist-induced AMPA receptor insertion, synaptic potentiation and fast-acting antidepressant responses[3,18]. Therefore, vti1a/VAMP7 DKD may alter postsynaptic efficacy via inhibition of eEF2 phosphorylation.

To explore whether vti1a/VAMP7 DKD alters phosphorylation levels of eEF2, we measured eEF2 phosphorylation in solubilized cultured mouse hippocampal neurons from eEF2K knockout (KO) mice or eEF2K wild-type (WT) littermate controls[33] with or without vti1a/VAMP7 DKD. Protein samples were subjected to immunoblotting using antibodies against total eEF2, phospho-eEF2 and Rab-GDI as a loading control. In eEF2K WT neurons, loss of vti1a and VAMP7 reduced phospho-eEF2 levels (Fig. 4a,b). Consistent with previous results, phospho-eEF2 was absent in cultures from eEF2K KO animals[3,33,35], demonstrating primary antibody specificity (Fig. 4a,b). Similarly, acute NMDAR block reduced phospho-eEF2 levels in control neurons while activity blockade with TTX did not, as shown previously[19], and chronic loss of vti1a and VAMP7 was as, if not more, effective at reducing eEF2 phosphorylation (Supplementary Fig. 4c,d). Taken together, these results suggest that spontaneous neurotransmission driven by vti1a- and VAMP7-containing vesicles activates NMDARs and maintains postsynaptic eEF2 in a phosphorylated state.

To further evaluate the dependence of synaptic scaling on eEF2K, we recorded AMPA-mEPSCs in cultured neurons from eEF2K KO or littermate WT mice with and without vti1a/VAMP7 DKD (Fig. 4c). Cumulative probability analysis of

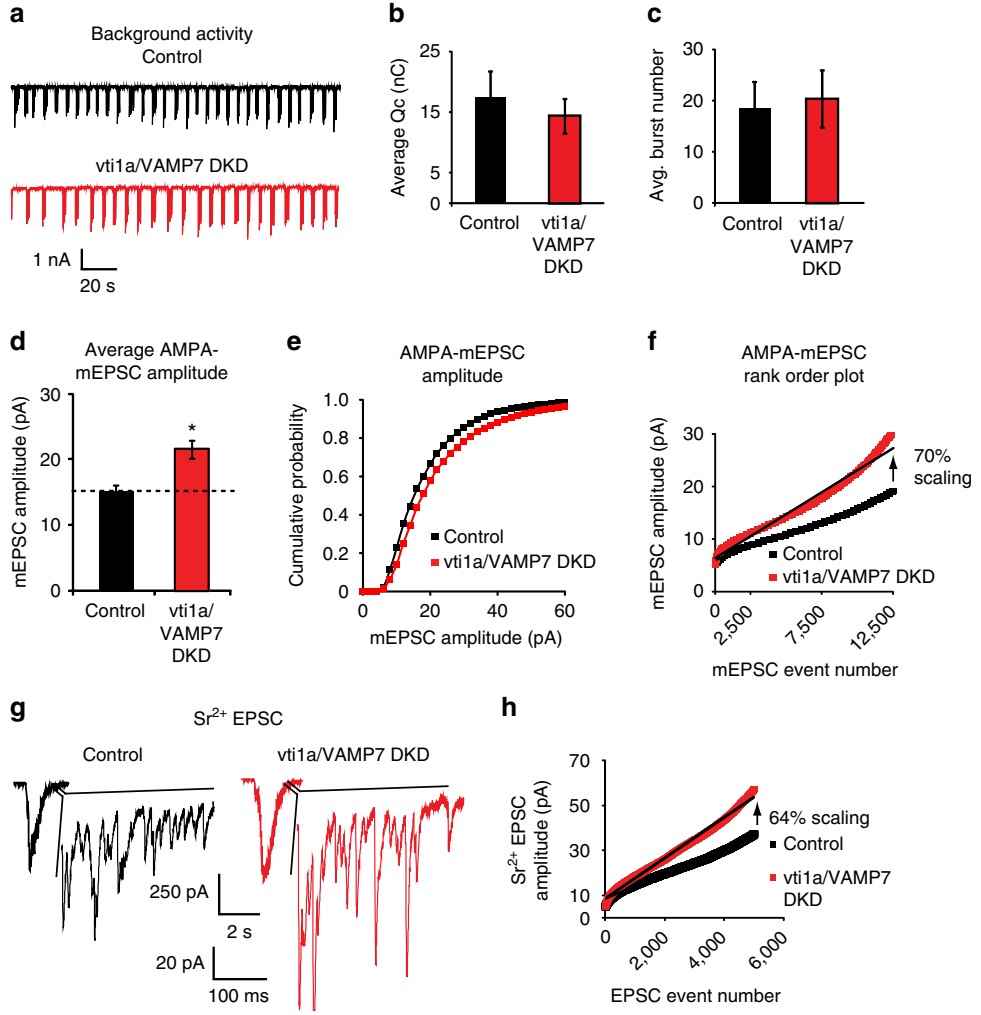

**Figure 3 | Vti1a and VAMP7 loss elicits scaling of unitary event amplitudes without changes in excitatory network activity.** (**a**) Representative traces showing background network activity recordings from control and vti1a/VAMP7 DKD neurons cultured from rat hippocampus. (**b**) No differences were seen in average total charge transfer between control and vti1a/VAMP7 DKD neurons from 3 min of recording (control: $n = 9$ neurons from three independent cultures; vti1a/VAMP7 DKD: $n = 10$ neurons from three independent cultures; $P = 0.55$). (**c**) No differences were seen in average burst number between control and vti1a/VAMP7 DKD neurons during background network activity analysed in **b** ($P = 0.82$). (**d**) Average AMPA-mEPSC amplitudes from vti1a/VAMP7 DKD neurons were significantly increased compared with control neurons from recordings analysed in Fig. 1d ($P = 0.0005$). (**e**) Cumulative probability histograms of AMPA-mEPSC amplitudes from recordings analysed in Fig. 1d (Kolmogorov–Smirnov test; $P = 0.0001$). (**f**) Rank-order plot of AMPA-mEPSC amplitudes from recordings analysed in Fig. 1d. The slopes of the linear fits indicate a multiplicative increase (70% scaling) in vti1a/VAMP7 DKD neurons. (**g**) Representative traces showing evoked asynchronous unitary EPSC recordings in the presence of $Sr^{2+}$ and absence of $Ca^{2+}$ in control and vti1a/VAMP7 DKD neurons. The top scale refers to the evoked bulk events and the bottom scale refers to the enlarged insets showing asynchronous unitary EPSCs. (**h**) Rank-order plot of asynchronous evoked unitary AMPA-EPSC amplitudes from control and vti1a/VAMP7 DKD neurons (control: $n = 5$ recordings from three independent cultures; vti1a/VAMP7 DKD: $n = 4$ recordings from two independent cultures). The slopes of the linear fits of the two curves indicate a multiplicative increase (64% scaling) of unitary event amplitudes in vti1a/VAMP7 DKD neurons compared with control neurons.

inter-event intervals revealed a robust rightward shift in the curves from both WT and eEF2K KO neurons after vti1a/VAMP7 DKD (Fig. 4d), confirming the previous results from rat neurons wherein loss of these proteins reduced AMPA-mEPSC frequency (Fig. 1d). We also observed that Vti1a/VAMP7 DKD in eEF2K WT mouse hippocampal cultures produced a significant increase in AMPA-mEPSC amplitude that was absent in eEF2K KO neurons (Fig. 4e). This is reminiscent of the inability of eEF2K KO neurons to scale AMPA-mEPSC amplitudes after NMDAR blockers are applied[28]. These data demonstrate that vti1a- and VAMP7-mediated spontaneous neurotransmission requires eEF2K to induce synaptic scaling. However, the lack of change in AMPA-mEPSC

amplitude between eEF2K WT and eEF2K KO neurons (Fig. 4e) suggests additional adaptive processes for maintaining basal postsynaptic efficacy are active in the absence of eEF2K signalling.

**Postsynaptic loss does not alter NMDAR-dependent plasticity.** Because vti1a[10,36,37] and VAMP7 (refs 37,38) have well-documented roles in the endosomal pathway, it is possible they could directly influence postsynaptic AMPA receptor trafficking to modify AMPA-mEPSC amplitudes. To evaluate a potential direct postsynaptic effect of vti1a/VAMP7 DKD on mEPSC amplitude, we performed sparse transfection of vti1a and VAMP7

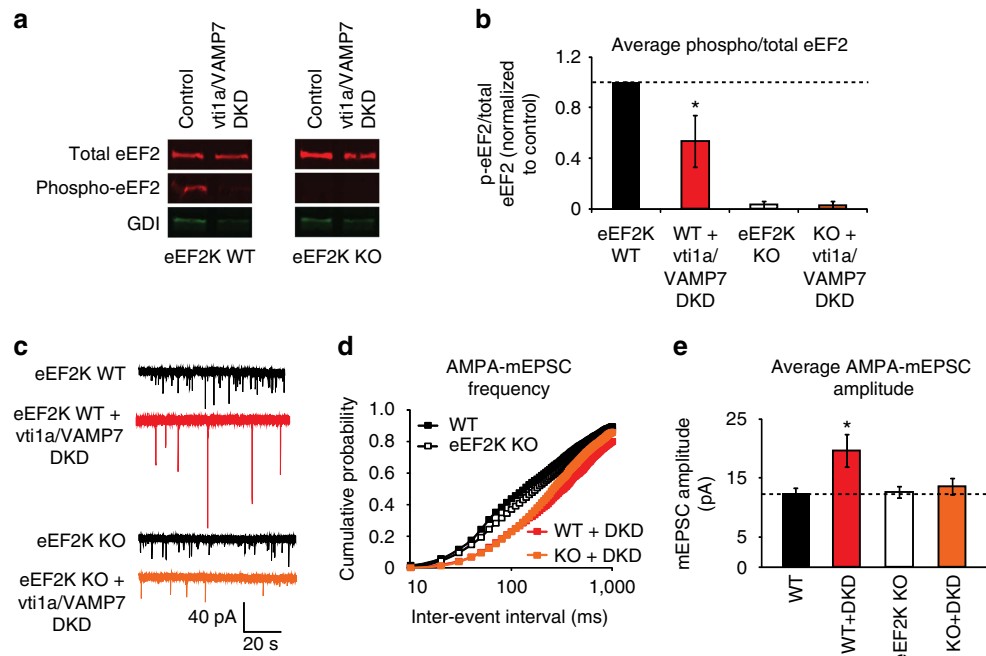

**Figure 4 | Synaptic scaling of AMPA-mEPSC amplitudes requires eEF2 kinase.** (**a**) Representative immunoblots showing total eEF2, phospho-eEF2 and loading control (GDI) levels in neuronal protein samples collected from WT and eEF2K KO sibling littermate neurons with or without vti1a/VAMP7 DKD. The break between genotypes represents the removal of irrelevant lanes from the full western blot image. (**b**) Quantitation of phospho-eEF2 levels compared with total eEF2 levels after normalization to the loading control ($n = 4$ independent cultures). Vti1a/VAMP7 DKD decreased phospho-eEF2 levels in WT neurons (corrected $P < 0.05$) while no phospho-eEF2 is observed in eEF2K KO neurons, as expected. (**c**) Representative traces of AMPA-mEPSC recordings in control or vti1a/VAMP7 DKD neurons from WT littermate or eEF2K KO mouse cultures. (**d**) Cumulative probability histograms of AMPA-mEPSC inter-event intervals (WT: $n = 12$ neurons from three independent cultures; WT + vti1a/VAMP7 DKD: $n = 6$ neurons from three independent cultures; eEF2K KO: $n = 13$ neurons from three independent cultures; eEF2K KO + vti1a/VAMP7 DKD: $n = 14$ neurons from four independent cultures). Vti1a/VAMP7 DKD significantly decreased the number of high-frequency AMPA-mEPSC events in both eEF2K WT and eEF2K KO neurons (Kolmogorov–Smirnov test; corrected $P < 0.001$, WT versus WT + vti1a/VAMP7 DKD; corrected $P < 0.001$, eEF2K KO versus eEF2K KO + vti1a/VAMP7 DKD). (**e**) Average AMPA-mEPSC amplitudes from data analysed in **d**. A significant increase in AMPA-mEPSC amplitude was seen in WT neurons with vti1a/VAMP7 DKD but not in eEF2K KO neurons (WT versus WT + vti1a/VAMP7 DKD, corrected $P < 0.05$; eEF2K KO versus eEF2K KO + vti1a/VAMP7 DKD, corrected $P > 0.05$).

DKD in single neurons as verified by soluble green fluorescent protein (GFP) fluorescence (Fig. 5) and immunocytochemistry. We then recorded AMPA-mEPSCs from transfected and non-transfected neurons, which both received predominantly non-transfected presynaptic inputs due to the sparse expression pattern ($\sim$5–20 transfected neurons per recording coverslip). No differences were found in AMPA-mEPSC amplitudes between non-transfected and vti1a/VAMP7 DKD-transfected neurons (Fig. 5d). Likewise, no differences were found in AMPA-mEPSC frequency (non-transfected: $2.86 \pm 0.88$ Hz, $n = 10$ neurons from three independent cultures; vti1a/VAMP7 DKD-transfected: $2.06 \pm 0.41$ Hz, $n = 11$ neurons from three independent cultures; $P = 0.41$). Altogether, these data suggest that vti1a and VAMP7 do not act via postsynaptic localization to modulate spontaneous release or produce synaptic scaling of AMPA-mEPSC amplitudes.

We next explored whether postsynaptic loss of vti1a and VAMP7 induces alterations in AMPA receptor function and basal neurotransmission in an intact circuit. To do so, we stereotaxically injected adeno-associated virus (AAV) driving shRNAs directed against vti1a and VAMP7, along with a GFP marker, into subregions of the hippocampus *in vivo* in adult mice (Fig. 6a,b; Supplementary Fig. 6). Loss of vti1a and VAMP7 in the CA1 subregion of the hippocampus did not alter input–output relationships of the Schaffer collateral pathway (Fig. 6c), suggesting that no change in AMPA receptor

function was observed in stimulus-dependent responses after reduction of postsynaptic vti1a and VAMP7 protein levels. Paired-pulse ratios were also unchanged (Fig. 6d), suggesting that vti1a/VAMP7 DKD in area CA1 does not retrogradely interfere with presynaptic function of the Schaffer collaterals arising from area CA3. Finally, to test whether postsynaptic loss of vti1a and VAMP7 impairs eEF2K-dependent synaptic plasticity, we measured synaptic potentiation induced by the NMDAR blocker ketamine, which is an age-dependent form of plasticity that requires eEF2K signalling and AMPA receptor insertion[3,39]. This form plasticity has also been previously associated with the rapid antidepressant responses elicited by ketamine[3,18,39]. After vti1a/VAMP7 DKD in area CA1, ketamine-induced synaptic potentiation of Schaffer collateral synapses is similar to control (Fig. 6e), supporting the culture data to suggest that the effects of vti1a and VAMP7 DKD on eEF2K-dependent synaptic function do not arise from alterations in receptor trafficking by postsynaptic organelles containing vti1a and VAMP7. Instead, changes in vti1a- and VAMP7-dependent spontaneous neurotransmission at presynaptic terminals may be responsible.

**Presynaptic loss impairs NMDAR-dependent plasticity.** We next examined the presynaptic effects of vti1a/VAMP7 DKD using the mouse hippocampal circuit. We employed an optical measurement of vesicle recycling to examine presynaptic function

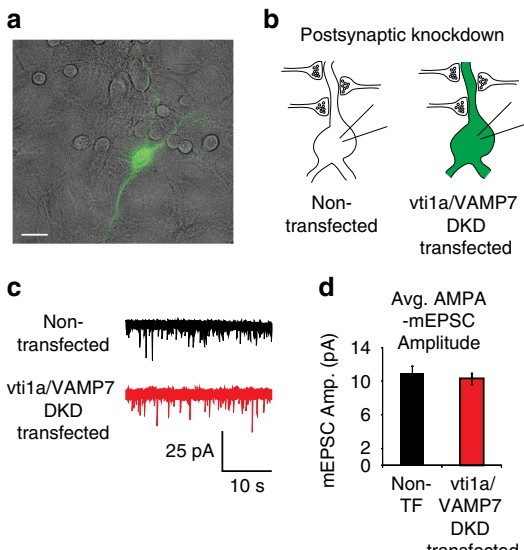

**Figure 5 | Postsynaptic loss of vti1a and VAMP7 is insufficient to elicit synaptic scaling of excitatory unitary event amplitudes.**
(**a**) Representative image of a GFP-positive neuron expressing shRNA constructs against vti1a and VAMP7 overlaid by a brightfield image showing many neighbouring GFP-negative (non-transfected control) neurons. Scale bar, 30 μm. (**b**) Diagram of experimental design to test the potential postsynaptic effects of vti1a/VAMP7 DKD in rat hippocampal cultures. AMPA-mEPSC recordings were made from both GFP-negative, control neurons and GFP-positive, vti1a/VAMP7 DKD-transfected neurons that receive predominantly control presynaptic inputs due to very low transfection efficiency. (**c**) Representative AMPA-mEPSC recordings from non-transfected (control) and vti1a/VAMP7 DKD-transfected neurons. (**d**) AMPA-mEPSC amplitudes did not differ between non-transfected and vti1a/VAMP7 DKD-transfected postsynaptic neurons (non-transfected: $n = 10$ recordings from three independent cultures; vti1a/VAMP7 DKD-transfected: $n = 11$ recordings from three independent cultures; $P = 0.56$).

due to the inability of electrophysiology to distinguish action potential-independent neurotransmission arising from individual, selectively infected presynaptic fibres. In the presence of TTX to block action potentials, we loaded presynaptic vesicles with an antibody against the luminal domain of synaptotagmin1 in 100 μm hippocampal slices after *in vivo* stereotaxic injection of AAV driving shRNAs against vti1a and VAMP7 into dentate gyrus (Supplementary Fig. 7a). We identified glutamatergic synapses in infected axons within area CA3 by co-staining with an antibody against vesicular glutamate transporter 1 (vGluT-1) and observed a reduction in synaptic levels of synaptotagmin 1 antibody loading in vti1a/VAMP7 DKD neurons, as shown by a leftward shift in the cumulative probability plot (Supplementary Fig. 7b). Staining intensity of vGluT-1 in the same synapses, however, did not change after vti1a/VAMP7 DKD (control: $86.5 \pm 4.2$ AU; vti1a/VAMP7 DKD: $78.7 \pm 3.0$ AU; $P = 0.14$). These data are consistent with our earlier findings in cultured neurons (Fig. 1a,b) and further show that spontaneous vesicle recycling is reduced after presynaptic reduction of vti1a and VAMP7 protein levels in an intact circuit.

To examine whether presynaptic vti1a/VAMP7 DKD impacts excitatory synaptic function in an intact circuit, we stereotaxically injected AAV driving shRNAs against vti1a and VAMP7 into area CA3 *in vivo* and performed field potential electrophysiology of Schaffer collateral synapses in acute slices (Fig. 7a,b).

In addition to better parsing out the effects of vti1a and VAMP7 location, this experiment avoids potential confounds in the cultured neuron model related to altered inhibitory spontaneous frequency (Supplementary Fig. 5)[40]. Similar to vti1a/VAMP7 DKD in area CA1, slices with DKD in area CA3 did not exhibit altered input–output relationships or paired-pulse ratios in Schaffer collateral synapses (Fig. 7c,d), suggesting that synchronous action potential-evoked neurotransmission at excitatory synapses is unaltered. However, in contrast to slices with vti1a/VAMP7 DKD in area CA1, slices with DKD in area CA3 exhibited impaired ketamine-induced synaptic potentiation (Fig. 7e), which supports the hypothesis that presynaptic vti1a and VAMP7 loss interferes with the NMDAR block-induced eEF2 signalling cascade responsible for this form of plasticity[3,18,39]. To determine whether this impairment extends to other forms of synaptic potentiation that have not been linked to dynamic changes in spontaneous neurotransmission, we also induced theta burst or high frequency stimulation long-term potentiation (LTP) in these slices. We found that both types of LTP are normal after presynaptic vti1a/VAMP7 DKD (Fig. 7f; Supplementary Fig. 8), suggesting that inhibition of vti1a- and VAMP7-dependent neurotransmission selectively interferes with NMDA blocker-induced synaptic plasticity.

## Discussion

Although previous work has proposed a link between spontaneous neurotransmission and the maintenance of glutamatergic synaptic efficacy[3,4], this premise has not been selectively tested in the absence of simultaneous alterations in excitatory action potential-evoked neurotransmission. Here we report that reducing protein levels of vti1a and VAMP7, two vesicular SNAREs found on spontaneously recycling vesicles, selectively inhibited spontaneous neurotransmission both in cultured neurons and in hippocampal slices. We also found that loss of vti1a and VAMP7 reduced multi-quantal bursts of AMPA-mEPSCs and subsequent resting NMDA receptor activation in cultured neurons, which led to a decrease in postsynaptic eEF2 phosphorylation. This reduction in spontaneous neurotransmission produced increased efficacy of AMPAR- but not GABAR-dependent signals, indicating neurotransmitter selectivity in downstream signalling cascades. Due to the multiplicative increase in AMPA-mEPSC amplitudes, this could be part of a homeostatic mechanism for setting basal excitatory synaptic strength. Interestingly, the selective presynaptic loss of vti1a and VAMP7 prevented acute NMDAR blocker-induced and eEF2K-dependent synaptic potentiation in slices. This inhibition of plasticity by vti1a and VAMP7 loss was specific, as LTP was intact. The similar results obtained in the culture model and the slice model of vti1a and VAMP7 loss suggests that this pathway may be important for controlling synaptic efficacy in both immature and mature neurons. Collectively, these results indicate that spontaneous neurotransmission can be regulated independently from action potential-evoked excitatory neurotransmission to affect specific postsynaptic signalling cascades and subsequent synaptic function.

Many presynaptic proteins differentially affect spontaneous and action potential-evoked neurotransmitter release, but few molecular manipulations of one form of neurotransmission leave the other unaltered[9]. Previous work, however, has indicated that synaptic vesicles containing either vti1a or VAMP7 selectively traffic at rest, with vti1a being a more specific marker of spontaneously recycling vesicles[15]. VAMP7 belongs to a resting pool of vesicles that can fuse spontaneously[14],

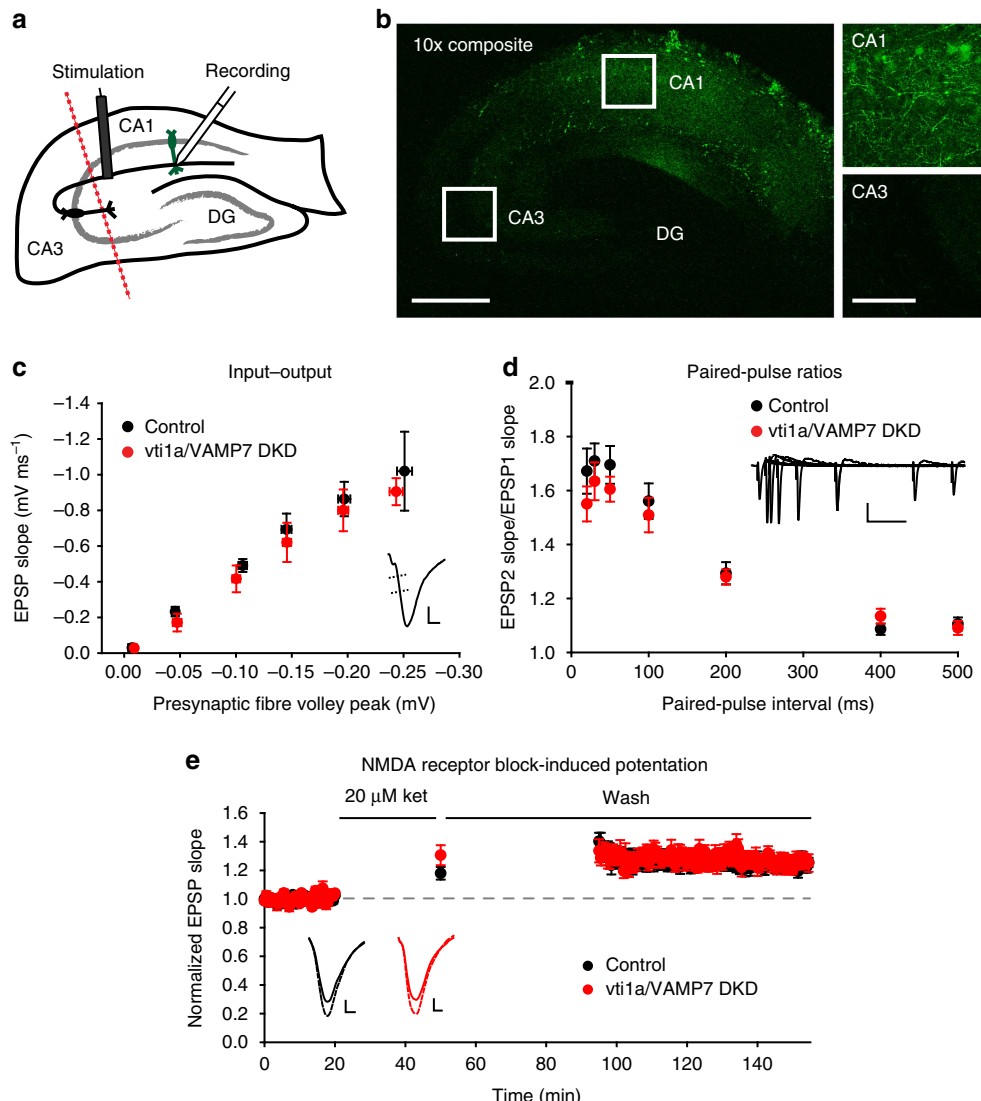

**Figure 6 | Vti1a and VAMP7 loss in hippocampal area CA1 is insufficient to alter neurotransmission in Schaffer collateral synapses. (a)** Diagram of the experimental design to test the effects of vti1a/VAMP7 DKD in area CA1 of mouse hippocampus. *In vivo* viral delivery to area CA1 of shRNA constructs against vti1a and VAMP7 with a GFP marker was followed by acute slice preparation and field potential electrophysiology of the Schaffer collateral synaptic pathway. Dotted line represents removal of area CA3 before recording. DG, dentate gyrus. **(b)** Left: composite image at 10 × magnification of GFP expression in a hippocampal slice made 3 weeks after injection of vti1a/VAMP7 DKD virus into area CA1. Scale bar, 500 µm. Right: 40 × magnification of areas indicated by the white boxes. Scale bar, 100 µm. **(c)** Input–output responses were not altered by vti1a/VAMP7 DKD (control: $n = 10$ slices from 7 mice; vti1a/VAMP7 DKD: $n = 12$ slices from 10 mice; control slope versus vti1a/VAMP7 DKD slope, $P = 0.99$). Inset: representative trace showing that the initial slope of the excitatory postsynaptic potential (EPSP) is measured against the amplitude of the presynaptic fibre volley. Scale is 0.3 mV by 2 ms. **(d)** Paired-pulse facilitation was not significantly different at 20, 30, 50, 100, 200, 400 or 500 ms inter-stimulus intervals after vti1a/VAMP7 DKD (control: $n = 14$ slices from 7 mice; vti1a/VAMP7 DKD: $n = 18$ slices from 11 mice; corrected $P > 0.05$ for all inter-stimulus intervals). Inset: representative traces showing responses at the various inter-stimulus intervals. Scale is 0.5 mV by 100 ms. **(e)** Ketamine-induced synaptic potentiation was not altered by vti1a/VAMP7 DKD (control: $n = 10$ slices from 7 mice; vti1a/VAMP7 DKD: $n = 6$ slices from 6 mice; $P = 0.77$). Field potential responses are normalized to the average baseline slope. Solid horizontal lines indicate application of 20 µM ketamine and then ACSF wash. Inset: representative traces from before (solid line) and after (dashed line) ketamine treatment. Scales are 0.2 mV (control) and 0.3 mV (vti1a/VAMP7 DKD) by 2 ms.

asynchronously under strong stimulation[14,16], or in response to the secreted glycoprotein Reelin[13]. Our goal was to selectively manipulate spontaneous neurotransmission via these proteins and target specific neuronal compartments to address whether distinct synaptic vesicle pools are capable of activating divergent postsynaptic signalling pathways. The maintenance of normal excitatory signalling in cultured neuronal networks and in Schaffer collateral synapses in slices after loss of vti1a and VAMP7 suggests that we achieved manipulation of spontaneous neurotransmission without appreciably modifying

action potential-evoked excitatory neurotransmission at the time points measured. Altogether, these results provide proof-of-principle that functionally distinct synaptic vesicle pools can be molecularly manipulated in a manner independent from other forms of neurotransmission.

We found that loss of vti1a and VAMP7 selectively alters AMPA receptor-dependent postsynaptic efficacy through eEF2 signalling despite depending on a quantitatively minor subset of spontaneous release events. The canonical vesicular SNARE synaptobrevin2 controls the majority of spontaneous

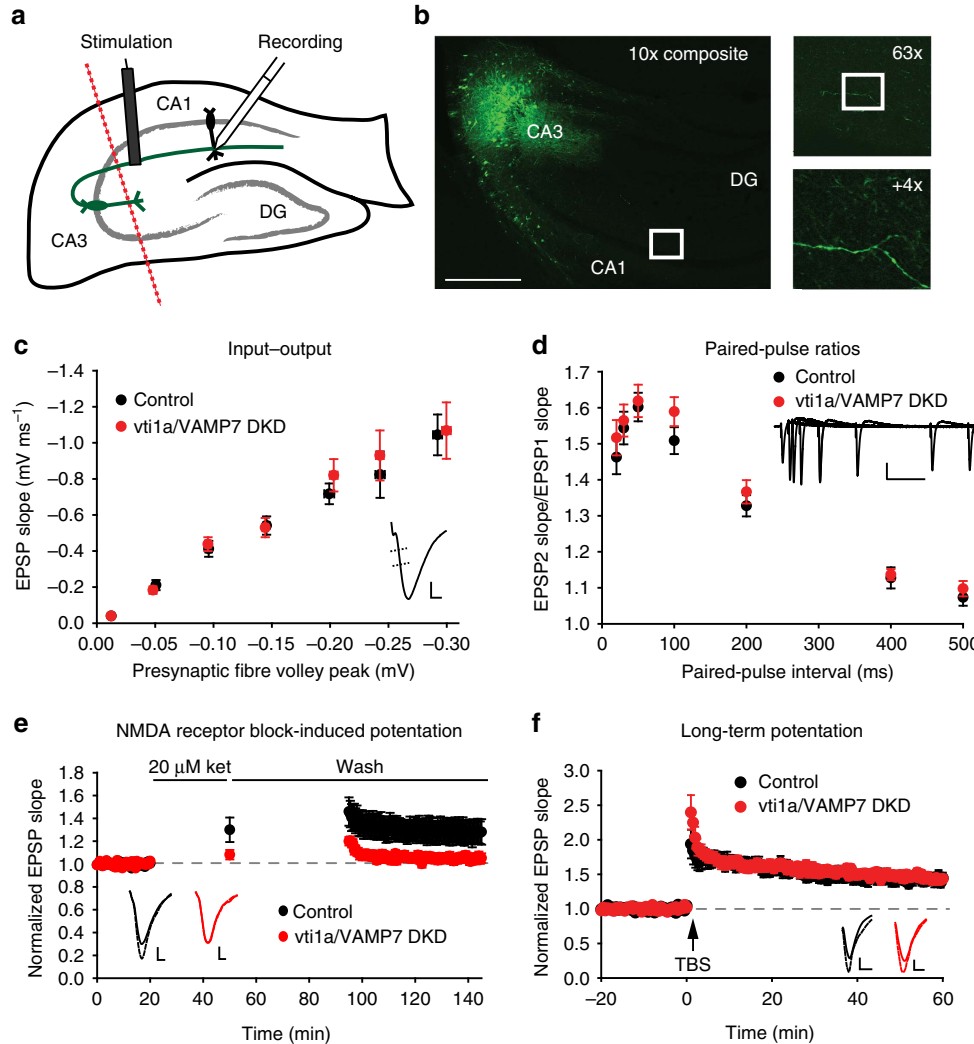

**Figure 7 | Presynaptic loss of vti1a and VAMP7 impairs NMDA receptor block-induced synaptic plasticity in Schaffer collateral synapses. (a)** Diagram of the experimental design to test the effects of vti1a/VAMP7 DKD in axons of mouse hippocampus. *In vivo* viral delivery to area CA3 of shRNA constructs against vti1a and VAMP7 with a GFP marker was followed by acute slice preparation and field potential electrophysiology of the Schaffer collateral synaptic pathway in CA1. Dotted line represents removal of area CA3 before recording. DG, dentate gyrus. **(b)** Composite image at $10 \times$ magnification of GFP expression in a hippocampal slice made 3 weeks after injection of vti1a/VAMP7 DKD virus into area CA3. Scale bar, 500 μm. Top inset: $63 \times$ magnification of the boxed region in the left image. GFP-positive Schaffer collateral axons can be observed. Bottom inset: $4 \times$ optical zoom of the boxed region in the top inset. **(c)** Input–output responses were not altered by presynaptic vti1a/VAMP7 DKD (control: $n = 25$ slices from 17 mice; vti1a/VAMP7 DKD: $n = 27$ slices from 18 mice; control slope versus vti1a/VAMP7 DKD slope, $P = 0.76$). Inset: representative trace showing that the initial response slope is measured against the amplitude of the presynaptic fibre volley. Scale is 0.4 mV by 2 ms. **(d)** Paired-pulse facilitation was not significantly different at 20, 30, 50, 100, 200, 400 or 500 ms inter-stimulus intervals after vti1a/VAMP7 DKD (control: $n = 37$ slices from 21 mice; vti1a/VAMP7 DKD: $n = 36$ slices from 20 mice; corrected $P > 0.05$ for all inter-stimulus intervals). Inset: representative traces showing responses at the various inter-stimulus intervals. Scale is 0.5 mV by 100 ms. **(e)** Ketamine-induced synaptic potentiation was reduced by presynaptic vti1a/VAMP7 DKD (control: $n = 25$ slices from 17 mice; vti1a/VAMP7 DKD: $n = 21$ slices from 14 mice; $P = 0.019$). Solid horizontal lines indicate application of 20 μM ketamine and then ACSF wash. Inset: representative traces from before (solid line) and after (dashed line) ketamine treatment. Scales are 0.3 mV (control) and 0.2 mV (vti1a/VAMP7 DKD) by 2 ms. **(f)** Long-term potentiation (LTP) was not altered by presynaptic vti1a/VAMP7 DKD (control: $n = 11$ slices from seven mice; vti1a/VAMP7 DKD: $n = 12$ slices from eight mice; $P = 0.69$). Arrow indicates when theta burst stimulation (TBS) was applied to Schaffer collateral axons. Inset: representative traces showing increased response (dashed line) after LTP induction. Scales are 0.3 mV (control) and 0.2 mV (vti1a/VAMP7 DKD) by 4 ms.

fusion events in addition to evoked neurotransmission, and yet its loss of function does not alter unitary postsynaptic efficacy[41]. To reconcile these findings, we tested whether vti1a- and VAMP7-containing vesicles preferentially activate postsynaptic NMDARs during spontaneous neurotransmission. Indeed, the present study uncovers a significant contribution of vti1a- and VAMP7-enriched vesicle pools to bursts of quantal release events that render resting NMDA receptor-mediated neurotransmission sensitive to the action of the use-dependent blocker MK-801. In addition to prior evidence showing that spontaneous neurotransmission-driven NMDAR activation can be reduced by use-dependent blockers even under physiological $Mg^{2+}$ concentrations[42], this control over postsynaptic $Ca^{2+}$ influx and subsequent $Ca^{2+}$ signalling provides a putative mechanism for how spontaneous neurotransmission alters postsynaptic signalling cascades[28].

Although it remains possible that release of other molecules besides glutamate, such as neuropeptides, could be affected by loss of vti1a and VAMP7, the resulting reduction in resting NMDAR signalling is a likely source of eEF2K deactivation, resulting in reduced eEF2 phosphorylation and de-suppression of dendritic protein translation[19]. As shown previously, eEF2 kinase KO synapses maintain basal neurotransmission, possibly through compensatory mechanisms due to the constitutive nature of the KO, but do not exhibit plasticity after loss of spontaneous neurotransmission[28]. It is possible, therefore, that dynamic phosphorylation of eEF2 is more important for this form of synaptic plasticity than basal phosphorylation levels. This effect may also be bidirectional, as we observed decreased AMPA-mEPSC amplitudes after vti1a overexpression, and previous work has shown that increasing vesicle release through α-latrotoxin increases phosphorylation of eEF2 in a manner dependent on NMDARs[19]. Consistent with the hypothesis that reduced NMDAR activation is the mechanism by which vti1a and VAMP7 loss induces scaling, pharmacological NMDA receptor block in cultured neurons did not further scale AMPA-mEPSC amplitudes in vti1a/VAMP7 DKD neurons. Future work will be necessary to explore how much vti1a and VAMP7 individually contribute to this signalling pathway and the subsequent functional changes, but the mild phenotypes of constitutive vti1a[43] and VAMP7 (ref. 44) KO mice may suggest functional redundancy between these proteins or other compensation mechanisms.

The increase of quantal amplitudes we observed in $Sr^{2+}$ after loss of vti1a and VAMP7 demonstrates that this manipulation not only enhances spontaneous AMPA-mEPSC amplitudes but also the amplitudes of quantal events that contribute to evoked neurotransmitter release, suggesting that spontaneous neurotransmission can influence the basal state of synapses involved in other forms of release. Interestingly, this scaling did not translate into detectable changes in network activity in cultured neurons or basal evoked neurotransmission in hippocampal slices, as would be expected from a homeostatic process. Homeostatic postsynaptic receptor scaling is classically proposed to occur in response to negative feedback from global changes in neuronal activity[45]. It is possible that activity levels in our culture and slice models are altered transiently after initial loss of protein and normalize before the time of assessment, but another intriguing hypothesis is that the signalling cascades driven by spontaneous neurotransmission and normally associated with homeostasis are, in fact, separable from signalling cascades driven by activity. Supporting this view, a recent study showed that chick embryonic spinal cord neurons appear to produce scaling in response to changes in spontaneous neurotransmission after, and sometimes in the absence of, homeostatic changes in evoked neurotransmission[46]. Although vti1a- and VAMP-dependent neurotransmission triggers signalling cascades known to be involved in homeostatic processes[19] and, therefore, may physiologically participate in activity homeostasis, it is possible that these forms of neurotransmission operate with some degree of independence[9]. It is clear, however, that these vesicular SNAREs markedly contribute to the functional state of synapses.

To parse out the role of vti1a and VAMP7 localization in these effects, we also employed hippocampal subregion-specific manipulations within an intact circuit to test presynaptic and postsynaptic effects of vti1a and VAMP7 loss. Although vti1a and VAMP7 have well-known roles in the trans-Golgi network and endosomes[10,36–38], which could potentially alter postsynaptic AMPA receptor trafficking, loss of vti1a and VAMP7 in pyramidal neurons and interneurons of CA1 did not appreciably alter Schaffer collateral synaptic function. Cultured neurons exhibited similar effects, in that AMPA-mEPSC frequency and amplitude were unaltered in postsynaptic neurons transfected with vti1a and VAMP7 DKD. As it is possible that inhibitory signalling interacts, via a change in GABA-mIPSC frequency, in dissociated cultures with spontaneous excitatory signalling[40], it was also vital to test in a more direct manner whether presynaptic manipulations in excitatory neurons explain the phenomenon. Specific presynaptic, but not postsynaptic, loss of vti1a and VAMP7 in the Schaffer collateral pathway did not alter basal neurotransmission after 3 weeks, suggesting that vti1a and VAMP7, regardless of subcellular location, do not appreciably alter basal action potential-evoked excitatory neurotransmission. However, presynaptic loss of vti1a and VAMP7 reduced NMDAR blocker-induced synaptic potentiation while postsynaptic loss did not. As we found that vti1a and VAMP7 drive a portion of resting NMDAR activation, it is likely that preemptive elimination of these spontaneous vesicle pools curtails further acute reductions in resting NMDAR activation by NMDAR open-channel blockers and, therefore, prevents dynamic dephosphorylation of eEF2. This effect appears similar to the prevention of additional eEF2 signalling-dependent AMPA-mEPSC scaling after NMDAR blockers were added to cultured neurons with reduced vti1a and VAMP7 levels. In contrast, LTP induction in hippocampal slices requires action potential stimulation and is normal after presynaptic loss of vti1a and VAMP7, suggesting that it relies on independent postsynaptic signalling cascades.

This dependence of NMDAR blocker-induced potentiation on spontaneous neurotransmission, eEF2 signalling, and AMPA receptor insertion is supported by previous work showing that potentiation is induced by depletion of neurotransmitter from vesicles by the vacuolar $H^+$-ATPase inhibitor folimycin and is absent in eEF2K KO mice[3]. Rather intriguingly, the same mechanism has been implicated in mediating the rapid antidepressant effects of ketamine in vivo[3,18]. This link between the release events facilitated by vti1a and VAMP7 and their potential impact on behavioural responses may have key implications for the development of novel therapeutic strategies targeted against specific forms of neurotransmission to treat neuropsychiatric disorders[47]. As action potential-driven neurotransmission is indispensable for nervous system function, strategies that selectively target spontaneous neurotransmission and its downstream signalling pathways may provide novel therapeutic targets with limited side effects in contrast to more global manipulations of neurotransmission.

Taken together, our findings demonstrate that selective presynaptic impairment of spontaneous release is sufficient to alter synaptic efficacy. Our data suggest a mechanism in which spontaneous neurotransmitter release mediated by a subset of the presynaptic machinery establishes unitary synaptic efficacy independent of global alterations in excitatory neuronal activity. The spontaneous release machinery itself is sensitive to $Ca^{2+}$ signalling, neuromodulators, and other signalling pathways ranging from endoplasmic reticulum stress to transcriptional regulation[6,48,49] and operates as a critical regulatory node to maintain synaptic efficacy and ensure the necessary dynamic range of synaptic responses. The subset of spontaneous neurotransmission controlled by vti1a and VAMP7, through apparent preferential activation of resting NMDARs and postsynaptic eEF2 signalling, is also poised for modulation by therapeutic agents to regulate synaptic plasticity.

## Methods

**Primary neuron culture.** Dissociated hippocampal cultures were prepared as previously described[50]. Briefly, whole hippocampi were dissected from male and female postnatal day 0–3 (P0–3) Sprague-Dawley rat pups or male and female P0-3 eukaryotic elongation factor 2 kinase (eEF2K) KO or WT littermate mouse pups (gift of Dr Alexey Ryazanov). Tissue was trypsinized (10 mg ml$^{-1}$ trypsin; Sigma-Aldrich, St Louis, MO) for 10 min at 37 °C, mechanically dissociated by pipetting, and plated on coverslips coated in Matrigel (BD Biosciences, Bedford, MA) for rat cultures or poly-D-lysine (Sigma-Aldrich) for mouse cultures. Cytosine arabinoside (4 μM; Sigma-Aldrich) was added at day in vitro (DIV) 1 and reduced to 2 μM at DIV 4. Cultures were maintained at 37 °C and 5% CO$_2$. All cultured neuron experiments were performed at DIV 14–21, and experiments were performed on multiple independent cultures.

**Lentiviral preparation.** We generated lentiviral expression constructs encoding anti-vti1a and anti-VAMP7 shRNAs by cloning hairpin sequences against versions of the proteins into pL307 (gift of Dr Thomas Südhof). The hairpin sequences used to knock down vti1a[15] and VAMP7 (ref. 13) expression have been previously described and validated. Our most effective knockdown constructs, vti1a-1.3 and VAMP7-KD3, resulted in an ~90% knockdown of vti1a and VAMP7 protein levels, respectively, and were used in this study (Supplementary Fig. 1). The construct expressing pHluorin-tagged vti1a used to rescue vti1a/VAMP7 DKD effects or for vti1a overexpression experiments has been previously described[15]. Recombinant viruses were prepared by transfection of human embryonic kidney (HEK) 293-T cells using Fugene 6 (Roche Applied Science, Indianapolis, IN) with the plasmids of interest together with plasmids encoding viral packaging and coating proteins (pRsv-Rev, pPRE-MALG and pVSVG). The virus was harvested from the HEK 293-T-conditioned medium 3 days post-transfection and added to the neuronal culture medium. Dissociated hippocampal cultures from rats or mice were infected with lentivirus at DIV 4. Using these techniques, we consistently obtained infection frequencies approaching 100%. Control neurons were obtained from the same cultures but not infected with lentiviruses or infected with the empty L307 vector expressing soluble GFP.

**Primary neuron transfection.** Primary rat hippocampal cultures were directly transfected at DIV 10 with knockdown constructs (vti1a-1.3 and VAMP7-KD3) to reduce levels of vti1a and VAMP7 in a small subset of neurons. Briefly, DNA constructs were mixed with Lipofectamine 2000 (Thermo Fisher Scientific, Waltham, MA) and incubated with neurons for 4 h at 37 °C and 5% CO$_2$. The DNA − Lipofectamine complexes were then removed and replaced with a mixture of fresh and conditioned neuronal culture medium. Cells expressing knockdown shRNA constructs were detected via soluble GFP co-expression at least 4 days after transfection. Images of transfected neurons were obtained at 512 × 512 pixels using a 40 × objective on a Nikon TE2000-U microscope (Nikon, Melville, NY) using NIS-Elements Advanced Research software (Nikon).

**Stereotaxic surgery.** Adult (8–13 weeks old) male C57Bl/6J mice were anaesthetized via intraperitoneal injection with 100 mg kg$^{-1}$ ketamine hydrochloride (Pfizer, New York, NY) and 10 mg kg$^{-1}$ xylazine (Vedco, Inc., St Joseph, MO). After mounting the mouse in the stereotaxic apparatus, a midline skin incision was made, and small holes were drilled into the skull above the target injection sites. Syringes (Hamilton, Reno, NV) with 33-gauge needles were guided to the target coordinates, and 1 μl AAV was delivered over 4 min with an additional 6 min allowance for diffusion before slowly withdrawing the syringes. Syringes were placed at a 10° angle, and bilateral coordinates relative to Bregma were as follows: CA3: − 1.9 mm anteroposterior, + 3.5 mm lateral, − 2.1 mm dorsoventral; CA1: − 1.9 mm anteroposterior, + 2.0 mm lateral, − 1.8 mm dorsoventral; and dentate gyrus: − 1.7 mm anteroposterior, + 1.7 mm lateral, − 2.0 mm dorsoventral. Before surgery and after recovery, mice were group-housed with a maximum of four siblings and kept on a 12-h light/dark cycle with ad libitum food and water. Mice were chosen at random within a cage to receive control or experimental AAV injections. All procedures were approved by the University of Texas Southwestern Medical Center's Institutional Animal Care and Use Committee.

**AAV preparation.** The hairpin sequences used to knock down vti1a and VAMP7 in the AAV vector were directed against the mouse versions of the proteins: vti1a, sense 5′-GAAGCAAATGGTTGCCAAT-3′; VAMP7, sense 5′-TAAGAGCCT AGACAAAGTG-3′. The genome plasmid for vti1a/VAMP7 DKD contained AAV2 inverted terminal repeats, an H1 promoter driving expression of shRNA against mouse vti1a, a CMV promoter driving expression of eGFP, a human growth hormone polyadenylation site, a U6 promoter driving expression of shRNA against mouse VAMP7, and a portion of the Woodchuck hepatitis virus posttranscriptional regulatory element as a spacer between the short interfering RNA hairpin and viral inverted terminal repeats (ITR). The control genome plasmid purchased from Cell Biolabs, Inc. (San Diego, CA) contained a CMV promoter driving expression of eGFP and a human growth hormone poly-adenylation site. AAV vectors of serotype DJ[51] were generated by the Stanford Neuroscience Gene Vector and Virus Core by transfecting AAV 293 cells (Agilent

Technologies, Santa Clara, CA) with the genome plasmid, pHELPER AAV helper plasmid (Agilent), and AAV rep-cap helper plasmid (pRC-DJ, gift of Dr Mark Kay) and then purifying the AAV using an iodixanol step gradient ultracentrifugation method followed by ultrafiltration. The genomic titer was determined by Q-PCR: control, 1.5 × 10$^{13}$ vg ml$^{-1}$; vti1a/VAMP7 DKD, 9.2 × 10$^{12}$ vg ml$^{-1}$. Efficient DKD of vti1a and VAMP7 protein levels in mouse hippocampus was observed via Western blot 3 weeks after stereotaxic injection (Supplementary Fig. 6); thus all experiments were performed at this time point. Injection sites were confirmed in each mouse via 30–100 μm brain sections made at the time of the slice preparation, and any hippocampi with mis-localized GFP were removed from the data set. To approximate infection efficiency, images from GFP and DKD-infected brain sections were intensity thresholded to include GFP-positive cell bodies using ImageJ software (NIH, Bethesda, MD). From these images, the percentage of CA1 or CA3 pyramidal layer area occupied by GFP-positive cell bodies was calculated from multiple slices for each hippocampus (CA1: 52.9 ± 3.6%, $n = 4$ hippocampi; CA3: 53.7 ± 11.5%, $n = 3$ hippocampi).

**Patch-clamp electrophysiology.** Whole-cell patch-clamp recordings were performed on hippocampal pyramidal neurons cultured in normal medium conditions in the presence of activity unless otherwise stated. Data were acquired using a MultiClamp 700B or AxoPatch 200B amplifier and Clampex 9.0 software (Molecular Devices, Sunnyvale, CA). Recordings were filtered at 2 kHz and sampled at 200 μs. A modified Tyrode's solution containing (in mM) 150 NaCl, 4 KCl, 2 MgCl$_2$, 2 CaCl$_2$, 10 glucose and 10 HEPES at pH = 7.4 was used as external bath solution. The pipette internal solution for all experiments except current-clamp contained (in mM): 115 Cs-MeSO$_3$, 10 CsCl, 5 NaCl, 10 HEPES, 0.6 EGTA, 20 Tetraethylammonium-Cl, 4 Mg-ATP and 0.3 Na$_3$GTP, and 10 QX-314 [N-(2,6-dimethylphenylcarbamoylmethyl)-triethylammonium bromide; EMD Millipore, Billerica, MA] at pH = 7.35 and 300 mOsm. For voltage-clamp experiments, all neurons were held at − 70 mV. The pipette internal solution for current-clamp experiments contained (in mM): 110 K-gluconate, 20 KCl, 10 NaCl, 10 HEPES, 4 Mg-ATP, 0.3 Na$_3$GTP and 0.6 EGTA at pH = 7.3 and 300 mOsm.

For isolation of miniature AMPA-EPSCs (AMPA-mEPSCs), 1 μM TTX (Enzo Life Sciences, Ann Arbor, MI), 50 μM picrotoxin (PTX), and 50 μM DL-2-Amino-5-phosphonopentanoic acid (AP-5; Abcam, Cambridge, MA) or 50 μM D-2-Amino-5-phosphonopentanoic acid (D-APV; Abcam, Cambridge, MA) were added to the bath solution. Bursts of AMPA-mEPSCs were defined as groups (>2) of consecutive events with inter-event intervals <40 ms. For isolation of miniature GABA-IPSCs (GABA-mIPSCs), 1 μM TTX, 50 μM AP-5 and 10 μM 6-cyano-7-nitroquinoxaline-2,3-dione disodium salt hydrate were added. For isolation of NMDA-mEPSCs, 1 μM TTX, 50 μM PTX, 10 μM 1,2,3,4-tetrahydro-6-nitro-2,3-dioxo-benzo[f]quinoxaline-7-sulfonamide disodium salt hydrate and 10 μM glycine were added while MgCl$_2$ was removed from the extracellular solution. NMDA-mEPSCs were recorded under continuous perfusion, and (5S,10R)-( + )-5-Methyl-10,11-dihydro-5H-dibenzo[a,d]cyclohepten-5,10-imine hydrogen maleate (MK-801) was used at 10 μM final concentration. AMPA-mEPSC and GABA-mIPSC recordings were 3 min in length, and Mini Analysis software (Synaptosoft, Fort Lee, NJ) was used to identify and analyse events. Cumulative probability plots of event frequency were prepared by combining histograms from individual recording experiments.

Background activity was recorded in voltage-clamp for 3 min with 50 μM PTX added to the bath solution, and total charge transfer was measured using Clampfit 9.0 software (Molecular Devices). Spontaneous action potential activity was recorded in current-clamp (without current injection) for 3 min in the absence of receptor blockers, and action potentials were detected using Clampfit 10.2 software (Molecular Devices) with a threshold amplitude of 0 mV. Action potential bursts were defined as depolarizations of membrane potential that produced at least two action potentials before returning to resting membrane potential. Evoked unitary EPSCs were elicited using field stimulation and recorded in Ca$^{2+}$-free bath solution with 50 μM AP-5, 50 μM PTX and 2 mM SrCl$_2$ added. For the bulk Sr$^{2+}$ response amplitude, at least three EPSC amplitudes were averaged for each neuron. Field stimulation was applied through parallel bipolar electrodes (FHC, St Bowdoin, ME) immersed in the perfusion chamber that delivered 34.5 mA pulses. A stimulation train of 60 action potentials at 0.2 Hz was used, and the amplitudes of the unitary events in 1-s periods at the end of the evoked response decay were measured using Mini Analysis software. Reagents were from Sigma-Aldrich unless otherwise stated.

**Brain slice preparation.** Mice were anaesthetized with isoflurane before decapitation and brain extraction. Brains were dissected to produce blocks containing the hippocampus and sliced at 100 or 400 μm thickness on a Leica VT 1000S vibratome (Leica Biosystems, Buffalo Grove, IL) in ice-cold oxygenated dissection buffer. Dissection buffer contained (in mM): 2.6 KCl, 1.25 sodium phosphate monobasic monohydrate, 26 sodium bicarbonate, 10 glucose, 0.5 CaCl$_2$, 5 MgCl$_2$ and 212 sucrose. Immediately after sectioning, area CA3 was surgically removed from 400 μM slices. Slices were then placed in oxygenated artificial cerebral spinal fluid (ACSF) containing (in mM): 124 NaCl, 5 KCl, 1.25 sodium phosphate monobasic monohydrate, 26 sodium bicarbonate, 10 glucose, 2 CaCl$_2$, and 1 MgCl$_2$. Slices were allowed to recover at 30 °C for at least 1 h for

100 µM slices and for at least 2 h for 400 µM slices before experiments. All reagents in dissection buffer and ACSF were from Sigma-Aldrich.

**Field potential electrophysiology.** Mouse hippocampal slices of 400 µm thickness were submerged in a recording chamber and perfused constantly at 3 ml min$^{-1}$ with oxygenated ACSF maintained at 30 °C. Excitatory postsynaptic field potentials at the Schaffer collateral synapse were elicited by 100 µs pulses delivered by a concentric bipolar stimulating electrode (FHC) and recorded by an ACSF-filled extracellular electrode placed in *stratum radiatum* of area CA1 within the hippocampus. Data were acquired using an Axoclamp 900A amplifier and Clampex 10.3 software (Molecular Devices). Recordings were filtered at 10 kHz and sampled at 50 µs. Data were analysed using Clampfit 10.2 software (Molecular Devices), and the experimenter was blinded to the experimental condition during recordings and analysis.

Responses were collected once every 30 s at 50–70% of the maximum response until a stable baseline was reached for at least 20 min. Any slices without a stable baseline were removed from the data set. For input–output experiments, a range of stimulation levels were applied that elicited between 0 and 100% of the maximum response. For paired-pulse recordings, the ratio of the initial response slope of the second field potential to that of the first field potential was measured. At least three responses were averaged at each stimulation level or inter-stimulus interval during input–output and paired-pulse protocols, respectively. For NMDA receptor block-induced potentiation and LTP experiments, response slopes were measured as a ratio to the average baseline response, and normalized data were averaged across experiments. Statistics were performed on the average normalized response of each slice during the last 20 min of recording. During NMDA receptor block-induced potentiation experiments, stimulation was halted while 20 µM S-(+)-ketamine hydrochloride (Sigma-Aldrich) was applied for 30 min, a test pulse was then elicited, ACSF was perfused for 45 min, and then stimulation resumed. LTP was induced via theta burst stimulation with the following parameters: three trains 10 s apart, each containing five bursts 200 ms apart with five pulses at 100 Hz within each burst. LTP was induced via high frequency stimulation with the following parameters: two trains 20 s apart, each containing one burst with 100 pulses at 100 Hz.

**Western blotting.** Sample preparation of cultured neurons was as previously described[49]. Immunoblotting to detect levels of total and phosphorylated eEF2 was performed according to the manufacturer's protocol (Li-Cor Odyssey Infrared Imaging System, Lincoln, NE). Membranes were incubated for at least 1 h at room temperature in blocking buffer (#927-40000, Li-Cor). Primary antibodies were incubated overnight at 4 °C and diluted in blocking buffer supplemented with 0.05% (v/v) Tween-20 (Surfact-Amps 20, #28320, Thermo Fisher). Primary antibodies used in these studies were directed against the following proteins and diluted as follows: anti-total eEF2 (1:750 dilution; #2331BC) and anti-phospho eEF2 (1:1,000 dilution; #2332BC) rabbit polyclonal antibodies (both from Cell Signaling Technologies, Danvers, MA) and anti-Rab-GDI mouse monoclonal antibody at 1:10,000 dilution (#130-011, Synaptic Systems, Goettingen, Germany). The membranes were washed three times for 10 min each with phosphate-buffered saline containing 0.05% (v/v) Tween-20 (PBS-T). IRDye-680-conjugated goat anti-rabbit (#926-32221, Li-Cor) and IRDye-800-conjugated goat anti-mouse secondary antibodies (#926-32210, Li-Cor) were diluted 1:10,000 in blocking buffer supplemented with 0.05% Tween-20 and 0.01% (v/v) sodium dodecyl sulfate and incubated with the membranes for 1 h at room temperature in the dark. The membranes were washed three times for 10 min each with PBS-T then once with PBS. Membranes were then scanned on a Li-Cor Odyssey machine using application software. Membranes were scanned at 169 µm resolution, high quality, and with intensity values of 3–6. Resulting images were exported as TIF files and integrated band intensities were quantified and exported to Microsoft Excel using the report feature.

Immunoblotting to assess vti1a and VAMP7 knockdown in cultured neurons was performed similarly to previously described[13,15]. For vti1a and VAMP7 knockdown in mouse hippocampi, protein lysates from whole hippocampus were prepared in RIPA buffer (pH = 7.4) containing 50 mM Tris (Thermo Fisher), 1.0% Igepal, 0.1% SDS, 0.5% sodium deoxycholate, 4 mM EDTA and 150 mM NaCl supplemented with 1 cOmplete Mini Tablet (#04-693-159-001; Roche) per 10 ml, 10 mM sodium pyrophosphate tetrabasic decahydrate, 50 mM NaF and 2 mM sodium orthovanadate (all from Sigma-Aldrich unless otherwise stated). Total protein concentration was quantified using Bradford protein assay. After SDS-polyacrylamide gel electrophoresis and transfer to membranes, membranes were incubated for at least 2 h in PBS-T with 5% milk protein. Primary antibodies were diluted in PBS-T with 5% milk and incubated overnight at 4 °C. Primary antibodies used were anti-vti1a mouse monoclonal antibody (1:500; #611220, BD Biosciences, San Jose, CA), anti-VAMP7 rabbit polyclonal antibody (1:1,000; #232-003, Synaptic Systems), and anti-Rab-GDI mouse monoclonal antibody (1:5,000). Membranes were washed 3× with PBS-T and incubated for 1 h in secondary antibodies diluted in PBS-T with 5% milk. Secondary antibodies used were horseradish peroxidase-conjugated goat anti-mouse (#55550) or anti-rabbit (#55676) at 1:5,000 (both from MP Biomedicals, Solon, OH). Membranes were washed again with PBS-T, and bands were developed using enzymatic chemiluminescence and exposure to films. Vti1a and VAMP7 levels

were normalized to GDI, and quantification was performed by an experimenter blinded to condition using ImageJ software. Images were cropped for presentation, but uncropped images are available in Supplementary Fig. 9. Antibodies were all commercially available and validated by the manufacturers.

**Luminal synaptotagmin 1 antibody uptake.** Cultured rat hippocampal neurons were pre-incubated in Tyrode's recording saline containing 1 µM TTX prepared as previously described for 5 min at room temperature and then incubated for 15 min at room temperature with mouse monoclonal antibodies against the luminal epitope of synaptotagmin 1 (1:100 dilution; Synaptic Systems) in Tyrode's solution containing 1 µM TTX. The neurons were then washed 4 times in PBS and fixed in 4% paraformaldehyde for 15 min at room temperature. Synaptotagmin 1 uptake was detected using Alexa-488-conjugated goat anti-mouse antibodies (Thermo Fisher Scientific). Negative controls were performed without the primary antibody in order to measure background fluorescence.

Mouse hippocampal slices of 100 µm thickness were treated with rabbit polyclonal antibodies against the luminal epitope of synaptotagmin 1 (1:100; Synaptic Systems) in oxygenated ACSF containing 1 µM TTX and maintained at 30 °C for 1 h. Slices were then washed 2× with ACSF and 1× with PBS before >30 min fixation with 4% paraformaldehyde at room temperature. Slices were then washed 1× with PBS containing 1% glycine and 4× with PBS before 2 h incubation at room temperature with PBS containing 20% goat serum (Sigma-Aldrich). To identify glutamatergic synapses, the slices were then treated for 2 days at 4 °C with guinea pig antibodies against vGluT-1 (1:2,000; EMD Millipore) in PBS containing 10% goat serum and 0.2% Triton-X-100 (Thermo Fisher Scientific). Synaptotagmin 1 uptake was then detected using Alexa-594-conjugated goat anti-rabbit antibodies (1:200; Thermo Fisher Scientific), and vGluT-1 was detected using Alexa-647-conjugated goat anti-guinea pig antibodies (1:500; Thermo Fisher Scientific). As with cultured neurons, negative controls were also performed with the synaptotagmin 1 primary antibody absent during the treatment.

Both stained coverslips and slices were imaged at 512 × 512 pixels on a Zeiss LSM510 META confocal microscope using a 63× objective and LSM 5 software (Zeiss, Okerkochen, Germany). Images of cultured neurons were acquired at 1× optical zoom while images of slices were acquired at 4× optical zoom. For cultured neurons, 15–30 puncta of constant diameter were selected for synaptotagmin 1 intensity analysis per image using LSM5 software. For synapse number counts in cultured neurons, synaptotagmin 1 puncta were automatically detected using ImageJ software. For slices, 15–20 vGluT-1-positive puncta that overlapped with GFP-positive mossy fibre axons in *stratum lucidum* of area CA3 were selected for analysis in each image using LSM 5 software before observing the synaptotagmin 1 channel by an experimenter blinded to the experimental condition. Background was measured in each slice from a region without vGluT-1 or synaptotagmin 1 staining. The absolute fluorescence intensities of synaptotagmin 1 loading were measured and calculated offline. Adjustments in brightness, contrast or colour of images made for the purposes of display in figures were uniformly applied to the entire image and across experimental conditions.

**Statistics.** Statistical analyses were performed using Excel 2010 (Microsoft, Redmond, WA) and SigmaPlot 12 (Systat Software, Inc. San Jose, CA) software. For two independent conditions of approximately normally distributed data with similar s.e.'s, an unpaired two-tailed *t*-test was used unless otherwise stated, while for non-normally distributed data a Mann–Whitney *U*-test was used unless otherwise stated. Statistical significance was defined as $P < 0.05$, and one-way ANOVA followed by Bonferroni correction for multiple comparisons was applied to determine significance in datasets with more than two groups. Grubb's test ($P < 0.01$) was used to determine outliers in slice electrophysiology experiments, which were then removed from the data set. Sample sizes were not statistically predetermined but conform to similar studies. All results are presented as mean ± s.e.m. unless otherwise stated.

**Data availability.** Data are available upon request to the corresponding author.

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

## Acknowledgements

We thank members of the Kavalali and Monteggia laboratories as well as Dr Helmut Kramer for insightful discussions and comments on the manuscript. We are grateful to Dr Manjot Bal for his insight at early stages of this project. This work was supported by NIH grants F32MH102915 (D.C.C.), R01MH066198 (E.T.K.), R01MH070727 (L.M.M.) as well as funding from the Brain & Behaviour Research Foundation (D.M.O.R., L.M.M. and E.T.K.) and the International Mental Health Research Organization (L.M.M.).

## Author contributions

D.C.C., D.M.O.R., L.M.M. and E.T.K. conceived and designed the study. D.C.C. and D.M.O.R. performed and analysed experiments. B.T. developed and validated viral constructs. D.C.C., D.M.O.R. and E.T.K. wrote the paper, and all authors critically evaluated the manuscript.
