## [Peer Review File · Nature Communications]

Reviewers' comments:

Reviewer #1 (Remarks to the Author):

Here the authors have sought to discriminate between the direct effects of evoked vs. spontaneous neurotransmitter release on postsynaptic efficacy at excitatory synapses in hippocampal neurons. The separation between the two modes of release was achieved by doubly knocking down vti1a and VAMP7, two of the SNAREs involved in endosomal trafficking that had been previously shown by the Kavalali group to be required for spontaneous neurotransmitter release. The authors report that double knock-down of vti1a and VAMP7 (DKD) decreases synaptic NMDA receptor activity and potentiates AMPA mEPSC amplitudes, and under the same condition, phosphorylated eEF2K is decreased. Moreover, the change in synaptic AMPA responses appears to be due to DKD of vti1a and VAMP7 in presynaptic but not postsynaptic neurons; DKD of presynaptic CA3 neurons in vivo results in block of synaptic potentiation induced by inhibiting NMDA receptors. Based on these findings, it is concluded that spontaneous release mediated by presynaptic vti1a and VAMP7 modulates postsynaptic AMPA responses by controlling NMDAR-dependent modulation of eEF2K activity. However, there are major concerns and technical weaknesses in experimental design, and the main conclusions are not well supported by the data shown.

Previously, the Kavalali group has demonstrated the involvement of spontaneous neurotransmitter release in synaptic plasticity, especially on the effects of acute block of spontaneous release on NMDAR-mediated synaptic potentiation. Moreover, others have shown that activation of NMDARs by spontaneous release serves to reduce functional synaptic AMPA receptors by suppressing eEF2K activity. In consideration of these prior reports, the present findings are not particularly novel and the conceptual advance is limited to an incremental one.

Major concerns:

- 1) One might expect vti1a/VAMP7 DKD to affect general endocytic traffic in postsynaptic neurons. In order to interpret the results to specific impairment of spontaneous release, one needs to specifically knock-down vti1a/VAMP7 in presynaptic neurons, not only in slice experiments but also in culture experiments. Alternatively, the lack of effect of postsynaptic DKD needs to be included as a control for all the experiments involving global DKD, not only for AMPA mEPSC amplitude shown in Fig 5.
- 2) It is of concern that DKD produces a striking decrease in mIPSC frequency, which is in stark contrast to the subtle decrease in mEPSC frequency. Given that cultures are infected with lentivirus at DIV 4, DKD neurons develop under conditions in which inhibitory inputs are greatly suppressed. As such, decreased inhibition in the culture network could also produce synaptic scaling that is secondary to compromising spontaneous release, and therefore, the chronic effects of DKD are difficult to interpret. Complications associated with chronic manipulation are also highlighted by the apparent compensation of mEPSC amplitude in eEF2K KO neurons that is not different from WT neurons.
- 3) The use of different experimental systems is problematic. There are differences in developmental stage in that slice experiments are from mature animals of at least 2 months in age whereas culture experiments are from 2 to 3 week old cultures. It is not clear if synaptic signaling mechanisms might differ between the two. Notably, experiments pertaining to NMDAR-dependent eEF2K signaling on synaptic AMPA receptor function have not been performed in slices.

Specific comments/questions (including points related to above):

- 4) Does vti1a/VAMP7 DKD-induced potentiation of AMPA mEPSC amplitude occlude synaptic scaling

induced by TTX-APV or APV (cf. Sutton et al., 2006)?

5) Fig 2: The rationale for including the washout step is not clear. What are the differences in the extent block of NMDAR-mEPSCs in the continued presence of MK-801 relative to the time before MK-801 application between control and DKD cultures? The weaker apparent block of NMDAR-mEPSC by MK-801 in DKD cultures seems to be due to a more potent recovery from MK-801. A plausible explanation could be faster diffusion of surface NMDAR under conditions of compromised exo-endocytic trafficking. Is the difference still observed in postsynaptic DKD neurons (cf. Fig. 5)?

6) Fig 3: A significant reduction in mIPSC frequency may have also contributed to the observed increase in AMPA mEPSC amplitude.

7) Supplementary Fig 2: "DKD+vti1a rescue" does not seem to simply rescue the decrease in mEPSC frequency but increases it. Does overexpression of vti1a in control neurons affect mEPSC frequency and amplitude?

8) Fig 4a,b: According to the authors' hypothesis, if NMDAR activity driven by vti1a/VAMP7 serves to phosphorylate eEF2K, then AP5 should have no additional effect on vti1a/VAMP7 DKD WT cultures. This should be tested. Also does AP5 treatment in WT control conditions decrease phosphorylated eEF2K?

9) Fig. 6: AAV infection of CA1 pyramidal cell layer is not convincing at all as shown. A higher magnification view should be included. Also, there seems to be substantial spread of the virus to area CA3. A lack of change in AMPA receptor function should be directly confirmed by monitoring quantal synaptic AMPA responses.

10) Fig. 7: Contrary to the statement in the text (p.11 line 267), in the images shown in panel a, VGluT1 staining appears to be brighter and larger in DKD neurons compared to control neurons. What type of neurons in which area of hippocampus is being examined here?

11) Fig. 8: As commented above, it is not clear the extent to which the observations shown here are related to findings from cultures (Fig. 1 to 5).

Reviewer #2 (Remarks to the Author):

In the study "Selective molecular impairment of spontaneous neurotransmission modulates synaptic efficacy" the authors make several highly interesting observations. They show that they can selectively reduce spontaneous glutamatergic neurotransmission by knocking down 2 SNARE proteins (VAMP7 & vti1a) while leaving evoked neurotransmission and spiking activity relatively intact. This is important as it represents an exceptionally specific perturbation. Strikingly, by simply reducing some spontaneous glutamatergic release they trigger a compensatory synaptic strengthening that appears to be a multiplicative synaptic scaling. The study extensively demonstrates this in cultured hippocampal cells in vitro and in vivo, and demonstrated that SNARE knockdown acts through a presynaptic mechanism. Further, the results suggest that NMDAergic spontaneous neurotransmission mediates this synaptic strengthening through effects on the phosphorylated state of eEF2. The study demonstrates distinct signaling pathways for evoked versus spontaneous neurotransmission and represents an important step forward for the field of homeostatic plasticity as it advances our understanding of both the underlying mechanisms and function of synaptic scaling. In addition it provides important insights to the actions of the antidepressant ketamine. The study is extensive and carried out in a technically sound manner.

My main concern with the study is understanding the scaling plasticity in the context of homeostatic control of spiking activity. The authors discuss scaling and homeostasis in the following manner - "scaling is classically proposed to occur in response to negative feedback from global changes in neuronal activity. Consistent with this premise, the change in synaptic efficacy following the loss of vti1a and VAMP7 did not alter network activity in cultured neurons or input-output and paired-pulse responses in brain slices, suggesting it does not permanently imbalance evoked excitatory signaling." I do not understand how scaling in this context relates to homeostasis. There seem to be 2 different general possibilities that I can think of, but I am not sure what the authors' perspective is on this. 1) It seems the authors are suggesting that their perturbation (the double knock down - DKD) never influences spiking activity, evoked release, probability of release, or number of inputs, but does trigger scaling. This would be interesting because it would appear to go against the general idea that scaling homeostatically controls spiking activity through compensatory adjustments in synaptic strength, at least for this form of scaling - rather scaling would be triggered by a reduction in spontaneous release independent of a spike feedback mechanism, and potentially even more striking is the finding that scaling occurred but had no influence on excitability (network activity, input-output responses). Do the authors believe that their form of scaling is only experienced at spontaneous synapses and not synapses mediating evoked transmission? - Fig 3F argues against this doesn't it? These topics need to be clarified. 2) Alternatively, do the authors believe that it is possible that this is a process that homeostatically controls spiking activity? In other words, the DKD reduces spiking shortly after transfection and scaling serves as a compensatory mechanism that brings activity levels back to control values, which is why they don't see changes in network activity or input-output responses 10+ days after the transfection. The DKD could influence spiking shortly after transfection by reducing a small proportion of evoked release, by reducing depolarizations due to spontaneous release, or some influence on the postsynaptic cell. Scaling would then act as a compensation to raise the excitability of the network. The problem with this possibility is that scaling should increase the input/output response, which didn't happen, or that the input-output response is maintained because some other synaptic characteristic is reduced (probability of release or number of inputs), but again this did not appear to occur. Please clarify how the reviewers reconcile these ideas.

Related to the above comments the authors write in the introduction concerning the observation that reducing spontaneous neurotransmission triggers scaling "this premise has never been selectively tested in the absence of simultaneous alterations in action potential-evoked neurotransmission". The authors do a very nice job of demonstrating that the vast majority of the effect of the DKD is on spontaneously released vesicles, and that this triggers scaling is consistent with previous work that shows the importance of spontaneous release (but when activity is also altered). However, can one say conclusively that neither vti1a nor VAMP7 have any effect on evoked release or spiking, and that this could contribute to plasticity? Have the authors looked at the onset of the transfection to see whether activity patterns/evoked release are perturbed - this would correspond to when homeostatic recovery from a perturbation would be occurring? Are there previous studies that have shown blockade of the spontaneous release had no influence on the activity within the cultured network?

For the authors to truly demonstrate synaptic scaling they should show this, by rank order plots with control amplitudes plotted vs DKD amplitudes, or even better using cumulative histograms scaled to the different conditions and then using a Kolmogorov-Smirnov test of the distributions as suggested by their cited reference (Kim Alger). The current methods shown do not demonstrate a multiplicative relationship between control and DKD amplitude distributions. Particularly important to show if the plasticity is occurring at all synapses as discussed above.

Minor concerns:

What was the actual reduction in average frequency in DKD and if bursts were removed from analysis was there still a reduction in mEPSC frequency? Although it is clear that faster ones were preferentially reduced it didn't seem to only be fast ones (see Figure 4d WT and WT/DKD). "Altogether, these results suggest that vti1a and VAMP7 drive high-frequency bursts of AMPA-mEPSCs", but this doesn't rule out the possibility that the DKD reduces the frequency of spontaneous events not associated with bursts, correct?

Figure 2 should be better explained - is block greater in controls because MK801 will preferentially reduce high frequency NMDA mini's?

Not clear why spiking activity is assessed only in the disinhibited culture. It would be more meaningful to test activity levels in control conditions for control and DKD cultures.

Please provide a fuller description of Sr+2 unitary events - what they represent.

Please provide a more detailed explanation of why the eEF2K KO, which removes eEF2 phosphorylation, doesn't trigger upscaling by itself (in absence of DKD) to the discussion -Why doesn't the eEF2K KO promote dendritic translation?

In the discussion it is a bit strange when the authors argue that CA1 infection does not lead to electrophysiological changes in input-output or paired-pulse response and this suggests the important changes are associated with DKD in the presynaptic population, but then show presynaptic CA1 DKD infection also shows no difference in input-output or paired-pulse response.

Reviewer #3 (Remarks to the Author):

Emerging evidence suggests that the efficacy of neural transmission is modified by changes in basal synaptic activity, governed in part by homeostatic up- or downregulation of signaling cascades coupled to neurotransmitter receptor regulation. In the Crawford et al., 2016 manuscript, the authors attempt to determine the contribution of spontaneous neurotransmission to the regulation of synaptic strength. Their results suggest that knocking down 2 non-canonical SNARE proteins (vti1a and VAMP7) in cultured neurons impaired spontaneous, high-frequency neurotransmitter release and this resulted in a decrease in the phosphorylation of eEFK2 (an elongation factor involved in translation regulation). Using a combination of hippocampal region selective DKD in knockout mice lacking eEF2, they go on to suggest that altered p-eEFK2 levels are the result of a presynaptic but not postsynaptic loss vti1a and VAMP7 which enhances compensatory synaptic scaling through an NMDA receptor-dependent mechanism. The combination of genetic, electrophysiology, molecular biology and imaging techniques is thorough and yields interesting insights into the role of the spontaneous component of synaptic transmission in mediating synaptic regulation. However, further evidence is required to solidify some of their conclusions particularly surrounding the mechanisms linking NMDARs to eEFK2 activity modulation and the pre- versus postsynaptic effects of DKD of vti1a and VAMP7.

Major Concerns:

1. As shown in many figures, such as Fig. 1c, e, Fig. 2c and Fig. 3, the amplitude of mEPSCs is clearly increased following vti1a/VAMP7 DKD. This would strongly suggest a postsynaptic scaling mechanism, likely involving an increase in the number of postsynaptic AMPARs. Then it is very difficult to understand why such a postsynaptic alteration is not translated to an alteration in the efficacy of evoked synaptic transmission. In fact, the data shown in Fig. 3f is more in line with a general increase

in synaptic efficacy, whether spontaneous or evoked.

2. Given that the authors suggest a role for NMDA receptors in the vti1a/VAMP7 DKD-induced effects on eEFK2, it is important to show that activating NMDARs prevents or reduces the effects of DKD on synaptic scaling and levels of eEFK2 activity. Application of NMDA to shRNA treated neurons and assaying for scaling or eEFK2 phosphorylation would bolster claims that NMDARs are a key component of the DKD effect and that this effect is through eEFK2.

3. In figure 6B, the localization of the GFP signal does not indicate subregion selectivity. The GFP signal appears highest in the perforant fiber pathway into the dentate gyrus and appears to be found in some CA3 pyramidal neurons which would be presynaptic. Moreover, the location of the cells expressing highest GFP signal is more consistent with interneuron subtypes in CA1 which are generally not the cell type reported to be the source of the effects reported throughout the manuscript.

4. Although the use of theta-burst stimulation to test the effects of vti1a and VAMP7 knockdown on evoked transmitter release may be more physiological than high-frequency stimulation (100Hz), it is well known that the enhancement observed during TBS is due in part, to disinhibition resulting from activation of GABAB autoreceptors which suppresses GABAergic transmission. Given the observed effects on mIPSC frequency, the lack of effect on TBS could be masked by the KD effects on inhibitory signaling. To bolster their claim of no effect on evoked transmission the authors should test a form of LTP that does not rely on disinhibition for its induction (1x100Hz, 1 sec).

5. Pg. 6, Top paragraph: The authors report that treatment with CdCl₂ results in no statistically different effect on AMPA-mEPSC burst frequency. However the frequency of mEPSC bursts is 1.13 Hz and 2.52 Hz (double the control) in the absence and presence of CdCl₂ respectively, with relatively high SEMs and relatively low n's per group which suggests that lack of power is contaminating the data. Either increase the n's or remove this statement from the manuscript as this data is not statistically powerful enough to draw conclusions.

6. Western blots shown in Fig. 4a are simply not acceptable. These samples need to be run on the same membrane, and the same membrane is needed to be sequentially probed for p-eEF2 and total eEF2 for a better comparison.

7. Generally, reporting of statistical methods and results is poor throughout the manuscript and legends which should be corrected.

Minor points:

1. The data would be clearer if the AMPA mEPSC amplitude graph (current Fig. 2 was moved to Fig. 1.). Additionally, for mPSC frequencies, the ordinal data reported (cumulative probability graphs) should be complemented by secondary graphs showing the average frequency per cell per group with appropriate statistical comparisons (t-tests).

2. In Fig. 3g,F the authors test whether evoked asynchronous release is changed following DKD of vti1a and VAMP7 and find a significant increase in EPSC amplitude. This is inconsistent with a strict effect on spontaneous release and also suggests the possibility that the effects they are observing are the result of changes in asynchronous release following APs in cultured neurons. This should be discussed further.

3. For the Ketamine-induced potentiation and TBS-LTP experiments, what measures of potentiation were compared? Were the LTP plots compared to baseline levels (relative potentiation), or were they

compared to each other post-potentialiation (control vs DKD)? Were cumulative time points (ie 20-30 min post stimulation) or single time points compared?

Response to Reviewers

“Selective Molecular Impairment of Spontaneous Neurotransmission Modulates Synaptic Efficacy”

Crawford DC, Ramirez DMO, Trauterman B, Monteggia LM, and Kavalali ET

We would like to thank the reviewers for their constructive evaluation of our manuscript and for the detailed comments provided. We believe we were able to address the points raised. In the last few months, we have performed new experiments and expanded the discussion in multiple sections of the manuscript.

The main issues addressed in the revised manuscript are:

1. We expanded the questions in the paper to include whether scaling up of excitatory synaptic events induced by loss of *vti1a* and VAMP7 prevents further scaling by known pharmacological methods, which we tested through new experiments (see Supplementary Fig. 4 and Results pp. 8 and 10).
2. We further probed the effects of *vti1a* overexpression on mEPSC frequency and amplitude with new experiments where we overexpressed *vti1a* in control neurons (see Supplementary Fig. 2 and Results p. 8).
3. We performed new current-clamp experiments to better determine whether endogenous activity in cultured neurons is altered in the absence of receptor blockers (see Results p. 8 and Supplementary Fig. 3).
4. We performed new experiments in slices from mice injected with the *vti1a* and VAMP7 double knockdown AAV in area CA3 to test whether non-inhibition-dependent, high frequency stimulation-induced long-term potentiation is altered (see Supplementary Fig. 7 and Results p. 14).
5. We performed new analyses and added new data about mEPSC burst frequencies (Results p. 6), mEPSC amplitudes (Fig. 3e), strontium EPSCs (Results p. 9), and CA1 GFP expression after AAV injection (Fig. 6b) that were requested by reviewers.
6. We clarified rationale and added more details about our experimental design in the NMDAR block experiments (see Results p. 7), strontium experiments (see Results p. 9), Western blot experiments (see figure legends for Fig. 4a and Supplementary Fig. 4c), synaptotagmin loading experiments (see Results p. 13 and the legend for Fig. 7a), and field potential recordings (see Methods p. 25) as requested by reviewers.
7. We expanded the discussion of the potential role of inhibition in the effects that we observe in mEPSCs after *vti1a* and VAMP7 loss (see Results p. 13 and Discussion p. 18).
8. We added an extended discussion about the potential relationship between *vti1a* and VAMP7 loss and homeostatic changes in action potential-driven activity to the manuscript (see Discussion pp. 17 and 18).
9. We addressed the comments regarding use of different model systems (acute slices from adult mice after *in vivo* loss of protein and cultured neurons from young mice after *in vitro* loss of protein) to study similar signaling cascades (see pp. 15, 17, and 18).

Our detailed responses to the reviewers are stated below:

Reviewer #1 (Remarks to the Author):

Here the authors have sought to discriminate between the direct effects of evoked vs. spontaneous neurotransmitter release on postsynaptic efficacy at excitatory synapses in hippocampal neurons. The separation between the two modes of release was achieved by doubly knocking down vti1a and VAMP7, two of the SNAREs involved in endosomal trafficking that had been previously shown by the Kavalali group to be required for spontaneous neurotransmitter release. The authors report that double knock-down of vti1a and VAMP7 (DKD) decreases synaptic NMDA receptor activity and potentiates AMPA mEPSC amplitudes, and under the same condition, phosphorylated eEF2K is decreased. Moreover, the change in synaptic AMPA responses appears to be due to DKD of vti1a and VAMP7 in presynaptic but not postsynaptic neurons; DKD of presynaptic CA3 neurons in vivo results in block of synaptic potentiation induced by inhibiting NMDA receptors. Based on these findings, it is concluded that spontaneous release mediated by presynaptic vti1a and VAMP7 modulates postsynaptic AMPA responses by controlling NMDAR-dependent modulation of eEF2K activity. However, there are major concerns and technical weaknesses in experimental design, and the main conclusions are not well supported by the data shown.

Previously, the Kavalali group has demonstrated the involvement of spontaneous neurotransmitter release in synaptic plasticity, especially on the effects of acute block of spontaneous release on NMDAR-mediated synaptic potentiation. Moreover, others have shown that activation of NMDARs by spontaneous release serves to reduce functional synaptic AMPA receptors by suppressing eEF2K activity. In consideration of these prior reports, the present findings are not particularly novel and the conceptual advance is limited to an incremental one.

Major concerns:

1) One might expect vti1a/VAMP7 DKD to affect general endocytic traffic in postsynaptic neurons. In order to interpret the results to specific impairment of spontaneous release, one needs to specifically knock-down vti1a/VAMP7 in presynaptic neurons, not only in slice experiments but also in culture experiments. Alternatively, the lack of effect of postsynaptic DKD needs to be included as a control for all the experiments involving global DKD, not only for AMPA mEPSC amplitude shown in Fig 5.

We agree that knocking down vti1a and VAMP7 selectively in presynaptic terminals of cultured neurons would be ideal, but in practice it is not feasible to isolate spontaneous electrophysiological responses from individual presynaptic neurons in a dissociated culture network. Due to the difficulties of isolating presynaptic and postsynaptic components in cultured neurons, we opted to perform slice experiments with infection in neuronal subpopulations to address this issue more directly. Also, as suggested by the reviewer, we have included frequency data from vti1a/VAMP7 DKD-transfected postsynaptic neurons in the manuscript (see Results p. 12).

2) It is of concern that DKD produces a striking decrease in mIPSC frequency, which is in stark contrast to the subtle decrease in mEPSC frequency. Given that cultures are infected with lentivirus at DIV 4, DKD neurons develop under conditions in which inhibitory inputs are greatly suppressed. As such, decreased inhibition in the culture

network could also produce synaptic scaling that is secondary to compromising spontaneous release, and therefore, the chronic effects of DKD are difficult to interpret. Complications associated with chronic manipulation are also highlighted by the apparent compensation of mEPSC amplitude in eEF2K KO neurons that is not different from WT neurons.

We know that changes in mIPSCs could have interdependent effects with mEPSCs (e.g. Wierda & Sorensen, 2014 *J Neurosci*). We could not exclude secondary changes or compensation effects in the culture model, which is another reason we chose to extend our study to the slice model where we focus on excitatory presynaptic effects of vti1a and VAMP7 loss. The presynaptic-specific studies in slices support the culture work, suggesting that the effects can occur at excitatory synapses independent of changes in inhibition. We thank the reviewer for giving us the opportunity to expand our discussion on this topic (see Results p. 13 and Discussion p. 18).

3) The use of different experimental systems is problematic. There are differences in developmental stage in that slice experiments are from mature animals of at least 2 months in age whereas culture experiments are from 2 to 3 week old cultures. It is not clear if synaptic signaling mechanisms might differ between the two. Notably, experiments pertaining to NMDAR-dependent eEF2K signaling on synaptic AMPA receptor function have not been performed in slices.

We respectfully disagree with the reviewer, as we consider the study of this signaling cascade in multiple systems to be a strength of the paper. The slice data mentioned by the reviewer were not included in the manuscript because they have been published previously. We have now bolstered explanations throughout the manuscript of relevant prior studies (e.g. Autry et al, 2011 *Nature*; Nosyreva et al, 2013 *J Neurosci*; Nosyreva et al, 2014 *Front Mol Neurosci*; see manuscript pages 9, 11, 12, and 17), which have collectively shown evidence that eEF2 signaling and increased AMPA receptor function are involved in NMDA receptor block-dependent synaptic potentiation in acute hippocampal slices from adult rodents. We feel that the similar results obtained in cultured neurons from young animals and in brain slices from adults support our conclusions, and we have added more about this to the Discussion (p. 15).

Specific comments/questions (including points related to above):

4) Does vti1a/VAMP7 DKD-induced potentiation of AMPA mEPSC amplitude occlude synaptic scaling induced by TTX-APV or APV (cf. Sutton et al., 2006)?

We thank the reviewer for this suggestion. We applied TTX with or without D-APV in control and vti1a/VAMP7 DKD cultures and found that there is no significant extra scaling in the vti1a/VAMP7 DKD+TTX/D-APV condition (see Results p. 8 and Supplementary Fig. 4). This experiment provides evidence that vti1a/VAMP7 DKD is likely inducing synaptic scaling via a redundant signaling cascade such as reduced NMDAR signaling.

5) Fig 2: The rationale for including the washout step is not clear. What are the differences in the extent block of NMDAR-mEPSCs in the continued presence of MK-801 relative to the time before MK-801 application between control and DKD cultures? The weaker apparent block of NMDAR-mEPSC by MK-801 in DKD cultures seems to be due to a more potent recovery from MK-801. A plausible explanation could be faster diffusion of surface NMDAR under conditions of compromised exo-endocytic trafficking. Is the difference still observed in postsynaptic DKD neurons (cf. Fig. 5)?

We washed out of MK-801 to prevent inclusion of any non-use-dependent block in our measurements. We agree that there is likely some recovery from MK-801 block due to receptor diffusion and other factors, but as shown in Huettner & Bean, 1988 (*PNAS*) and Tovar & Westbrook, 2002 (*Neuron*), this recovery is very slow (on the order of hours, not just a few minutes), especially in the absence of exogenously applied receptor agonists, and is likely not substantially contributing to the effects we see after 1-3 minutes of wash. We have added additional discussion of this to the Results (p. 7).

6) Fig 3: A significant reduction in mIPSC frequency may have also contributed to the observed increase in AMPA mEPSC amplitude.

Please see our response to point 2 above.

7) Supplementary Fig 2: "DKD+vti1a rescue" does not seem to simply rescue the decrease in mEPSC frequency but increases it. Does overexpression of vti1a in control neurons affect mEPSC frequency and amplitude?

In a new set of experiments, we found that vti1a overexpression in control neurons significantly increased mEPSC frequency and significantly decreased mEPSC amplitude. This has been added to Supplementary Figure 2 and the Results (p. 5).

8) Fig 4a,b: According to the authors' hypothesis, if NMDAR activity driven by vti1a/VAMP7 serves to phosphorylate eEF2K, then AP5 should have no additional effect on vti1a/VAMP7 DKD WT cultures. This should be tested. Also does AP5 treatment in WT control conditions decrease phosphorylated eEF2K?

We now show that phospho-eEF2 levels do, indeed, decrease after AP5 (See Supplementary Fig. 4c, d), which is consistent with prior work by Sutton et al, 2007 (*Neuron*).

9) Fig. 6: AAV infection of CA1 pyramidal cell layer is not convincing at all as shown. A higher magnification view should be included. Also, there seems to be substantial spread of the virus to area CA3. A lack of change in AMPA receptor function should be directly confirmed by monitoring quantal synaptic AMPA responses.

A new image with higher magnification is now shown in Figure 6. No hippocampi in which slices expressed GFP in the pyramidal CA3 layer were included in the CA1 DKD dataset. We have also discussed in the Results (p.12) why patch-clamp measurements of mEPSCs are not informative in the slice experiments, as we are only infecting a subset of presynaptic neurons.

10) Fig. 7: Contrary to the statement in the text (p.11 line 267), in the images shown in panel a, VGlut1 staining appears to be brighter and larger in DKD neurons compared to control neurons. What type of neurons in which area of hippocampus is being examined here?

These are mossy fibers arising from dentate gyrus, chosen for the dense axonal tract and the ability to distinguish individual boutons within individual axons. We only analyzed boutons that coincided with axonal GFP in order to focus on fibers without vti1a and VAMP7, and the intensity of vGluT-1 in these boutons did not significantly differ. We now have details in the figure legend and the Results (p. 13).

11) Fig. 8: As commented above, it is not clear the extent to which the observations shown here are related to findings from cultures (Fig. 1 to 5).

Vti1a/VAMP7 DKD in both cultured neurons and slices appears to reduce spontaneous neurotransmission, thereby inducing NMDA receptor-dependent and eEF2-dependent signaling cascades that insert postsynaptic AMPA receptors. This is why we believe additional acute induction of this signaling cascade in slices, as shown in Figure 8e, is prevented. Clarification has been added to the Discussion (pp. 15, 17, and 18).

Reviewer #2 (Remarks to the Author):

In the study "Selective molecular impairment of spontaneous neurotransmission modulates synaptic efficacy" the authors make several highly interesting observations. They show that they can selectively reduce spontaneous glutamatergic neurotransmission by knocking down 2 SNARE proteins (VAMP7 & vti1a) while leaving evoked neurotransmission and spiking activity relatively intact. This is important as it represents an exceptionally specific perturbation. Strikingly, by simply reducing some spontaneous glutamatergic release they trigger a compensatory synaptic strengthening that appears to be a multiplicative synaptic scaling. The study extensively demonstrates this in cultured hippocampal cells in vitro and in vivo, and demonstrated that SNARE knockdown acts through a presynaptic mechanism. Further, the results suggest that NMDAergic spontaneous neurotransmission mediates this synaptic strengthening through effects on the phosphorylated state of eEF2. The study demonstrates distinct signaling pathways for evoked versus spontaneous neurotransmission and represents an important step forward for the field of homeostatic plasticity as it advances our understanding of both the underlying mechanisms and function of synaptic scaling. In addition it provides important insights to the actions of the antidepressant ketamine. The study is extensive and carried out in a technically sound manner.

My main concern with the study is understanding the scaling plasticity in the context of homeostatic control of spiking activity. The authors discuss scaling and homeostasis in the following manner - "scaling is classically proposed to occur in response to negative feedback from global changes in neuronal activity. Consistent with this premise, the change in synaptic efficacy following the loss of vti1a and VAMP7 did not alter network activity in cultured neurons or input-output and paired-pulse responses in brain slices, suggesting it does not permanently imbalance evoked excitatory signaling." I do not understand how scaling in this context relates to homeostasis. There seem to be 2 different general possibilities that I can think of, but I am not sure what the authors' perspective is on this. 1) It seems the authors are suggesting that their perturbation (the double knock down - DKD) never influences spiking activity, evoked release, probability of release, or number of inputs, but does trigger scaling. This is would be interesting because it would appear to go against the general idea that scaling homeostatically controls spiking activity through compensatory adjustments in synaptic strength, at least for this form of scaling - rather scaling would be triggered by a reduction in spontaneous release independent of a spike feedback mechanism, and potentially even more striking is the finding that scaling occurred but had no influence on excitability (network activity, input-output responses). Do the authors believe that their form of scaling is only experienced at spontaneous synapses and not synapses mediating evoked transmission? - Fig 3F argues against this doesn't it? These topics need to be clarified. 2) Alternatively, do the authors believe that it is possible that this is a process that homeostatically controls spiking activity? In other words, the DKD reduces spiking

shortly after transfection and scaling serves as a compensatory mechanism that brings activity levels back to control values, which is why they don't see changes in network activity or input-output responses 10+ days after the transfection. The DKD could influence spiking shortly after transfection by reducing a small proportion of evoked release, by reducing depolarizations due to spontaneous release, or some influence on the postsynaptic cell. Scaling would then act as a compensation to raise the excitability of the network. The problem with this possibility is that scaling should increase the input/output response, which didn't happen, or that the input-output response is maintained because some other synaptic characteristic is reduced (probability of release or number of inputs), but again this did not appear to occur. Please clarify how the reviewers reconcile these ideas.

Related to the above comments the authors write in the introduction concerning the observation that reducing spontaneous neurotransmission triggers scaling "this premise has never been selectively tested in the absence of simultaneous alterations in action potential-evoked neurotransmission". The authors do a very nice job of demonstrating that the vast majority of the effect of the DKD is on spontaneously released vesicles, and that this triggers scaling is consistent with previous work that shows the importance of spontaneous release (but when activity is also altered). However, can one say conclusively that neither *vti1a* nor VAMP7 have any effect on evoked release or spiking, and that this could contribute to plasticity? Have the authors looked at the onset of the transfection to see whether activity patterns/evoked release are perturbed - this would correspond to when homeostatic recovery from a perturbation would be occurring? Are there previous studies that have shown blockade of the spontaneous release had no influence on the activity within the cultured network?

We cannot exclude the possibility that activity is transiently altered but homeostatically normalized over the time period before we record from the neurons. Unfortunately, this hypothesis is difficult to test because action potential-evoked activity is not mature (and therefore measurable) at the young age of lentiviral infection in cultures, and significant *vti1a* and VAMP7 knockdown could not be confirmed prior to three weeks post-injection in the slice experiments. We thank the reviewer for this thoughtful comment and have added additional discussion about this point to the manuscript (see Discussion pp. 17 and 18).

For the authors to truly demonstrate synaptic scaling they should show this, by rank order plots with control amplitudes plotted vs DKD amplitudes, or even better using cumulative histograms scaled to the different conditions and then using a Kolmogorov-Smirnov test of the distributions as suggested by their cited reference (Kim Alger). The current methods shown do not demonstrate a multiplicative relationship between control and DKD amplitude distributions. Particularly important to show if the plasticity is occurring at all synapses as discussed above.

Figure 3 now contains both a cumulative probability distribution and a rank order plot in addition to the average mEPSC amplitude.

Minor concerns:

What was the actual reduction in average frequency in DKD and if bursts were removed from analysis was there still a reduction in mEPSC frequency? Although it is clear that faster ones were preferentially reduced it didn't seem to only be fast ones (see Figure 4d WT and WT/DKD). "Altogether, these results suggest that *vti1a* and VAMP7 drive high-

frequency bursts of AMPA-mEPSCs", but this doesn't rule out the possibility that the DKD reduces the frequency of spontaneous events not associated with bursts, correct?

We have now analyzed the burst data and non-burst data separately, and we see a significant decrease in the frequency of burst-associated AMPA-mEPSCs but not non-burst-associated AMPA-mEPSCs after loss of *vti1a* and VAMP7 (see Results p. 6).

Figure 2 should be better explained - is block greater in controls because MK801 will preferentially reduce high frequency NMDA mini's?

We do believe that this result is consistent with the idea that control neurons contain more high-frequency mEPSCs than *vti1a*/VAMP7 DKD neurons, as MK-801 is a use-dependent blocker and should, therefore, have stronger block where there is stronger NMDAR activation (like when there are bursts of mEPSCs). Clarification has been added to the Results p. 7.

Not clear why spiking activity is assessed only in the disinhibited culture. It would be more meaningful to test activity levels in control conditions for control and DKD cultures.

We initially recorded excitatory network activity due to the overall focus on excitatory neurotransmission in the manuscript as well as the inability to distinguish excitatory and inhibitory currents in the conditions under which the neurons were recorded. We have since performed new current-clamp experiments to measure spontaneous action potential activity in the absence of receptor blockers and found no significant differences in action potential frequency, burst frequency, or number of action potentials per burst (see Results p. 8 and Supplementary Fig. 3).

Please provide a fuller description of Sr+2 unitary events - what they represent.

Additional descriptions of the strontium unitary events have been added to the Results p. 9 and the Discussion p. 17.

Please provide a more detailed explanation of why the eEF2K KO, which removes eEF2 phosphorylation, doesn't trigger upscaling by itself (in absence of DKD) to the discussion -Why doesn't the eEF2K KO promote dendritic translation?

The lack of basal effect in eEF2K KO mice and prevention of pharmacology-induced scaling has been shown previously (Park et al, 2008 *Neuron*; Nosyreva et al, 2013 *JNeurosci*; Reese & Kavalali, 2015 *eLIFE*). Total eEF2 levels are maintained in the kinase knockouts, and other pathways may developmentally compensate for the constitutive loss of eEF2 phosphorylation to maintain basal neurotransmission. We hypothesize that the dynamic changes in phosphorylation levels of eEF2 alter synaptic efficacy but may not be required for the development of basal neurotransmission. We have now included a discussion about this on p. 17.

In the discussion it is a bit strange when the authors argue that CA1 infection does not lead to electrophysiological changes in input-output or paired-pulse response and this suggests the important changes are associated with DKD in the presynaptic population, but then show presynaptic CA1 DKD infection also shows no difference in input-output or paired-pulse response.

We used the postsynaptic (CA1) manipulation to test whether non-presynaptic functions of *vti1a*

and VAMP7 were contributing to the observed phenotype, and the lack thereof was confirmed with NMDAR blocker-induced potentiation, which requires similar signaling cascades as those identified in the cultured neuron preparation. The presynaptic (CA3) manipulation tested whether presynaptic vesicles containing vti1a and VAMP7 alter synaptic phenotypes, and we found a deficit in this form of synaptic plasticity. The input-output and paired-pulse experiments show that, regardless of location, vti1a and VAMP7 do not appreciably disrupt basal action potential-evoked neurotransmission. We have clarified more in the Results (p. 12) and Discussion (p. 18).

Reviewer #3 (Remarks to the Author):

Emerging evidence suggests that the efficacy of neural transmission is modified by changes in basal synaptic activity, governed in part by homeostatic up- or downregulation of signaling cascades coupled to neurotransmitter receptor regulation. In the Crawford et al., 2016 manuscript, the authors attempt to determine the contribution of spontaneous neurotransmission to the regulation of synaptic strength. Their results suggest that knocking down 2 non-canonical SNARE proteins (vti1a and VAMP7) in cultured neurons impaired spontaneous, high-frequency neurotransmitter release and this resulted in a decrease in the phosphorylation of eEFK2 (an elongation factor involved in translation regulation). Using a combination of hippocampal region selective DKD in knockout mice lacking eEF2, they go on to suggest that altered peEFK2 levels are the result of a presynaptic but not postsynaptic loss vti1a and VAMP7 which enhances compensatory synaptic scaling through an NMDA receptor-dependent mechanism. The combination of genetic, electrophysiology, molecular biology and imaging techniques is thorough and yields interesting insights into the role of the spontaneous component of synaptic transmission in mediating synaptic regulation. However, further evidence is required to solidify some of their conclusions particularly surrounding the mechanisms linking NMDARs to eEFK2 activity modulation and the pre-versus postsynaptic effects of DKD of vti1a and VAMP7.

Major Concerns:

1. As shown in many figures, such as Fig. 1c, e, Fig. 2c and Fig. 3, the amplitude of mEPSCs is clearly increased following vti1a/VAMP7 DKD. This would strongly suggest a postsynaptic scaling mechanism, likely involving an increase in the number of postsynaptic AMPARs. Then it is very difficult to understand why such a postsynaptic alteration is not translated to an alteration in the efficacy of evoked synaptic transmission. In fact, the data shown in Fig. 3f is more in line with a general increase in synaptic efficacy, whether spontaneous or evoked.

Stimulus-evoked EPSC responses, especially in dissociated cultures after de-synchronization with strontium, are variable, but we did see a potential increase in strontium EPSC amplitude after loss of vti1a and VAMP7 (see new data in Results p. 9). We did not see this increase in the slice experiments when measuring synchronous stimulation-evoked responses (see Fig. 8) or in network activity in cultured neurons (see Results p. 8, Fig. 3, and new data in Supplementary Fig. 3).

2. Given that the authors suggest a role for NMDA receptors in the vti1a/VAMP7 DKD-induced effects on eEFK2, it is important to show that activating NMDARs prevents or reduces the effects of DKD on synaptic scaling and levels of eEFK2 activity. Application of NMDA to shRNA treated neurons and assaying for scaling or eEFK2 phosphorylation

would bolster claims that NMDARs are a key component of the DKD effect and that this effect is through eEFK2.

NMDA is likely to activate extra-synaptic receptors, not just synaptic receptors, making the manipulation nonspecific and likely to cause excitotoxic damage after hours of treatment. Instead, we have collected new data in which overexpression of vti1a not only increases the frequency of AMPA-mEPSCs but also scales AMPA-mEPSC amplitudes downward (see Supplementary Fig. 2), supporting our hypothesis. Similarly, Sutton et al, 2007 (*Neuron*) used α -latrotoxin to increase vesicle release and subsequently observed increased phosphorylation of eEF2 in a manner that was blocked by the NMDA receptor antagonist APV, suggesting that this pathway is involved. We have added additional discussion about this topic to the manuscript (see Discussion p. 17).

3. In figure 6B, the localization of the GFP signal does not indicate subregion selectivity. The GFP signal appears highest in the perforant fiber pathway into the dentate gyrus and appears to be found in some CA3 pyramidal neurons which would be presynaptic. Moreover, the location of the cells expressing highest GFP signal is more consistent with interneuron subtypes in CA1 which are generally not the cell type reported to be the source of the effects reported throughout the manuscript.

We did not include any slices with detectable GFP expression in the CA3 pyramidal layer in the CA1 DKD dataset. Higher magnification of the CA1 and CA3 areas of an example slice has now been added to Figure 6 to aid clarification. We expect both inhibitory and excitatory neurons to be infected in area CA1 but measure primarily excitatory synaptic function after Schaffer collateral stimulation in slices throughout the manuscript.

4. Although the use of theta-burst stimulation to test the effects of vti1a and VAMP7 knockdown on evoked transmitter release may be more physiological than high-frequency stimulation (100Hz), it is well known that the enhancement observed during TBS is due in part, to disinhibition resulting from activation of GABAB autoreceptors which suppresses GABAergic transmission. Given the observed effects on mIPSC frequency, the lack of effect on TBS could be masked by the KD effects on inhibitory signaling. To bolster their claim of no effect on evoked transmission the authors should test a form of LTP that does not rely on disinhibition for its induction (1x100Hz, 1 sec).

We thank the reviewer for this suggestion. We have now performed high frequency (100Hz for 1s) stimulation-induced LTP experiments after injection of virus into area CA3 and see no difference between control and vti1a/VAMP7 DKD slices. These data have been added to the manuscript (see Results p. 13 and Supplementary Figure 7).

5. Pg. 6, Top paragraph: The authors report that treatment with CdCl₂ results in no statistically different effect on AMPA-mEPSC burst frequency. However the frequency of mEPSC bursts is 1.13 Hz and 2.52 Hz (double the control) in the absence and presence of CdCl₂ respectively, with relatively high SEMs and relatively low n's per group which suggests that lack of power is contaminating the data. Either increase the n's or remove this statement from the manuscript as this data is not statistically powerful enough to draw conclusions.

This statement has been removed from the manuscript.

6. Western blots shown in Fig. 4a are simply not acceptable. These samples need to be

run on the same membrane, and the same membrane is needed to be sequentially probed for peEF2 and total eEF2 for a better comparison.

The eEF2K WT and eEF2K KO samples were run on the same membrane but their lanes were not next to each other. We apologize for the confusion and have added clarification to the figure legend.

7. Generally, reporting of statistical methods and results is poor throughout the manuscript and legends which should be corrected.

We followed journal guidelines when writing the paper and reporting statistics. Please also refer to the Checklist, which contains statistical details, for clarification.

Minor points:

1. The data would be clearer if the AMPA mEPSC amplitude graph (current Fig. 2 was moved to Fig. 1.).

As the interpretation of AMPA-mEPSC frequency and amplitude data from cultured neurons diverge in the narrative, we opted to organize the figures so that we could first discuss the changes in frequency (Fig. 1 and Fig. 2) before we discussed the changes in amplitude (Fig. 3-5).

Additionally, for mPSC frequencies, the ordinal data reported (cumulative probability graphs) should be complemented by secondary graphs showing the average frequency per cell per group with appropriate statistical comparisons (t-tests).

Because of the selective reduction of AMPA-mEPSC frequency within bursts, the difference between control and *vti1a*/VAMP7 DKD conditions is not detected via average frequency statistics. These data have been added to the Results (p. 5).

2. In Fig. 3g,F the authors test whether evoked asynchronous release is changed following DKD of *vti1a* and VAMP7 and find a significant increase in EPSC amplitude. This is inconsistent with a strict effect on spontaneous release and also suggests the possibility that the effects they are observing are the result of changes in asynchronous release following APs in cultured neurons. This should be discussed further.

Additional discussion of the strontium unitary events has been added to the Results p. 9 and the Discussion p. 17.

3. For the Ketamine-induced potentiation and TBS-LTP experiments, what measures of potentiation were compared? Were the LTP plots compared to baseline levels (relative potentiation), or were they compared to each other post-potentiation (control vs DKD)? Were cumulative time points (ie 20-30 min post stimulation) or single time points compared?

The normalized responses over the last 20 min of recording (40-60 min after starting post-plasticity measurements) were averaged for each slice, and statistics were performed on these values. We have added additional clarification to the Methods section (p. 25).

Reviewers' comments:

Reviewer #1 (Remarks to the Author):

The revised manuscript has been significantly improved with additional experiments and careful editing of the text. There are some remaining issues concerning the slice experiments in Figures 6-8.

Figure 7. The quantification of SytI uptake relative to vGlut1 puncta as shown is not convincing at all. The problem is that even for the controls, the SytI uptake signal is very weak with high background. Also, the synapse being studied is different from the CA3-CA1 synapses in Figures 6 and 8. I recommend moving this figure to supplementary.

Figure 6 and 8. The representative image shown for AAV infection targeting the area CA1 has been improved with limited spreading to CA3. However, the efficiency of CA1 knock-down, which is particularly crucial because the result shown is a negative one, remains unclear. What percentage of CA1 pyramidal neurons has been infected with the virus? Similarly, for the CA3 knock-down experiments, the high resolution image shows a rather isolated CA3 axon. Since field stimulation will sample axons originating from both non-infected and infected CA3 cells, one needs to provide estimates of the likely relative contributions to the EPSPs by quantifying the infection efficiency of CA3 pyramidal cells.

Reviewer #2 (Remarks to the Author):

In the study "Selective molecular impairment of spontaneous neurotransmission modulates synaptic efficacy" the authors make several highly interesting observations. The study was already fairly extensive and carried out in a technically sound manner (mEPSC freq + ampl, mIPSC freq + ampl, evoked currents, primary cultures and slices, appropriate controls). They have now made several significant additions to the study based on the reviewer's comments - increased vti1a expression produces a downscaling so plasticity is bidirectional, selective reduction of multiquantal bursts, show DKD induced scaling was occluded by NMDA/TTX treatment, new slice transfection experiments. The study has been significantly improved both in content and clarity. The authors have addressed all of my main concerns. I have just a few additional minor concerns that the authors should consider.

- 1) Figure 2C is still confusing because the equation is percent unblocked, but the y-axis is % blocked.
- 2) Describe how GABAergic minis were isolated – held at 0mv or patch electrodes were chloride loaded?
- 3) To claim that mIPSCs scaled the authors should show cumulative histogram.

Reviewer #3 (Remarks to the Author):

In the revised manuscript, the authors have adequately addressed most concerns raised during the previous round of reviewing, and it can now be recommended for publication in the journal as it is.

Response to Reviewers

“Selective Molecular Impairment of Spontaneous Neurotransmission Modulates Synaptic Efficacy”

Crawford DC, Ramirez DMO, Trauterman B, Monteggia LM, and Kavalali ET

We would again like to thank the reviewers for their constructive evaluation of our manuscript. We have addressed the points raised by the most recent round of review by clarifying sections of the manuscript and performing new image analyses.

Our detailed responses are below:

Reviewer #1 (Remarks to the Author):

The revised manuscript has been significantly improved with additional experiments and careful editing of the text. There are some remaining issues concerning the slice experiments in Figures 6-8.

Figure 7. The quantification of Syt1 uptake relative to vGlut1 puncta as shown is not convincing at all. The problem is that even for the controls, the Syt1 uptake signal is very weak with high background. Also, the synapse being studied is different from the CA3-CA1 synapses in Figures 6 and 8. I recommend moving this figure to supplementary.

This figure is now Supplementary Figure 7.

Figure 6 and 8. The representative image shown for AAV infection targeting the area CA1 has been improved with limited spreading to CA3. However, the efficiency of CA1 knock-down, which is particularly crucial because the result shown is a negative one, remains unclear. What percentage of CA1 pyramidal neurons has been infected with the virus? Similarly, for the CA3 knock-down experiments, the high resolution image shows a rather isolated CA3 axon. Since field stimulation will sample axons originating from both non-infected and infected CA3 cells, one needs to provide estimates of the likely relative contributions to the EPSPs by quantifying the infection efficiency of CA3 pyramidal cells.

It is difficult to determine an exact percentage of infected neurons within these experiments, but we have re-analyzed images to determine the percentage of CA1 and CA3 pyramidal layers occupied by GFP-positive cell bodies using intensity thresholding. With this method, approximately 50% of the pyramidal layer area in both regions contains GFP-positive cell bodies (see Methods p. 24).

Reviewer #2 (Remarks to the Author):

In the study “Selective molecular impairment of spontaneous neurotransmission modulates synaptic efficacy” the authors make several highly interesting observations. The study was already fairly extensive and carried out in a technically sound manner (mEPSC freq + ampl, mIPSC freq + ampl, evoked currents, primary cultures and slices, appropriate controls). They have now made several significant additions to the study based on the reviewer’s comments - increased vti1a expression produces a downscaling so plasticity is bidirectional, selective reduction of multiquantal bursts, show DKD induced scaling was occluded by NMDA/TTX treatment, new slice transfection experiments. The study has been significantly improved both in content and clarity. The authors have addressed all of my main concerns. I have just a few additional minor

concerns that the authors should consider.

1) Figure 2C is still confusing because the equation is percent unblocked, but the y-axis is % blocked.

The equation shows that the remaining activity after MK-801 is subtracted from the baseline activity, which determines the amount of activity blocked by MK-801, and then divided this number by the baseline activity to determine the percentage block. We have clarified this further in the Results (p. 7).

2) Describe how GABAergic minis were isolated – held at 0mv or patch electrodes were chloride loaded?

The patch electrodes contain chloride, and we held the cells at -70 mV. We used pharmacology (TTX, CNQX, AP-5) to isolate GABAergic minis. This has been clarified in the Methods (p. 24-25).

3) To claim that mIPSCs scaled the authors should show cumulative histogram.

We do not claim that the mIPSC amplitudes scale; we only claim that the frequency is decreased. We have clarified this in the sections that discuss potentially decreased inhibition in the system (see Results p. 13 and Discussion p. 18).

Reviewer #3 (Remarks to the Author):

In the revised manuscript, the authors have adequately addressed most concerns raised during the previous round of reviewing, and it can now be recommended for publication in the journal as it is.

REVIEWERS' COMMENTS:

Reviewer #1 (Remarks to the Author):

The authors have addressed my remaining concerns. I support the publication of the manuscript as is.

Reviewer #2 (Remarks to the Author):

My minor concerns have been addressed.